# Dynamic Service Fee Pricing under Strategic Behavior: Actions as Instruments and Phase Transition

**Rui Ai**
MIT
ruiai@mit.edu

**David Simchi-Levi**
MIT
dslevi@mit.edu

**Feng Zhu**
MIT
fengzhu@mit.edu

## Abstract

We study a dynamic pricing problem for third-party platform service fees under strategic, far-sighted customers. In each time period, the platform sets a service fee based on historical data, observes the resulting transaction quantities, and collects revenue. The platform also monitors equilibrium prices influenced by both demand and supply. The objective is to maximize total revenue over a time horizon $T$. Our problem incorporates three practical challenges: (a) initially, the platform lacks knowledge of the demand side beforehand, necessitating a balance between exploring (learning the demand curve) and exploiting (maximizing revenue) simultaneously; (b) since only equilibrium prices and quantities are observable, traditional Ordinary Least Squares (OLS) estimators would be biased and inconsistent; (c) buyers are rational and strategic, seeking to maximize their consumer surplus and potentially misrepresenting their preferences. To address these challenges, we propose novel algorithmic solutions. Our approach involves: (i) a carefully designed active randomness injection to balance exploration and exploitation effectively; (ii) using non-i.i.d. actions as instrumental variables (IV) to consistently estimate demand; (iii) a low-switching cost design that promotes nearly truthful buyer behavior. We show an expected regret bound of $\widetilde{\mathcal{O}}(\sqrt{T} \wedge \sigma_S^{-2})$ and demonstrate its optimality, up to logarithmic factors, with respect to both the time horizon $T$ and the randomness in supply $\sigma_S$. Despite its simplicity, our model offers valuable insights into the use of actions as estimation instruments, the benefits of low-switching pricing policies in mitigating strategic buyer behavior, and the role of supply randomness in facilitating exploration which leads to a phase transition of policy performance.

## 1 Introduction

A large number of transactions nowadays take place on third-party platforms, such as shopping on Amazon, taking rides on Uber, and ordering takeout food on DoorDash [33, 39]. For these platforms, deciding how to set the service fee is an important issue [35, 40, 49] that not only affects the platform's short-term revenue but also impacts long-term user retention. Therefore, understanding buyers' demand curve is of significant importance for platforms aiming to maximize their revenue.

**Example 1.1.** As an illustrative example, as a ride-hailing platform, Uber possesses information on the supply side (drivers), specifically the number of drivers available on the road at any given moment. Simultaneously, when buyers (passengers) request a ride, it matches drivers with passengers and charges a certain fee. Uber charges a booking fee for each reservation. Over a certain time period,

Uber's booking fee remains approximately the same, as shown in Figure 1. However, over a longer period of time, an exploratory dynamic pricing strategy can be beneficial to acquire more demand information.

Meanwhile, consumers often refrain from taking rides when prices are high even if these prices are within their willingness to pay, aiming at inducing Uber to reduce the price or offer them coupons (which can be understood as a negative service fee) to reduce future purchase expenses. This phenomenon is common in both the psychology [6] and the economics [27, 42] literature.

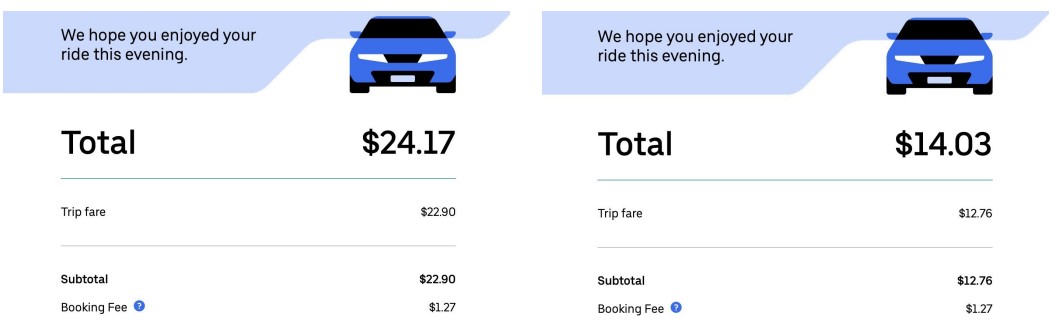

Figure 1: *Uber.* Above are two Uber rides from the same city in February 2024. The price of the two rides differs by nearly double, yet the booking fee remains at $1.27 for both, suggesting that a fixed booking fee mechanism might be employed in February. However, over a longer period of time, Uber may switch to a dynamic pricing strategy.

For third-party platforms, pricing service fees can face three challenges:

1. *Demand information needs to be learned.* Typically platforms can observe the number and quoted prices of sellers on the platform, i.e., the supply curve, but the preference of buyers, or the demand curve, is not observable. At the same time, due to legal restrictions [58], platforms in many cases cannot personalize pricing for different buyers but can only set a uniform price for a group of buyers. Therefore, it is crucial to learn about the buyer group's willingness to pay.

2. *Only equilibria can be observed.* Regarding buyer information, platforms can only observe the equilibrium price and quantity, say $P^e$ and $Q^e$ respectively, which depend on the service fee set by the platform itself and the changing supply curve. Thus, due to changes in the demand curve, $(P^e, Q^e)$ may only reveal partial information about the demand curve and thus can fail to recover the full demand curve. In the absence of randomness in the supply curve, $(P^e, Q^e)$ could even form an upward-sloping curve, far from the characteristics of a demand curve.

3. *Buyers may exhibit strategic behavior.* When buyers interact with the platform over an extended period, they may present a false demand curve to the platform, hoping to gain more benefits in future purchases, as shown in Example 1.1. The strategic behavior of buyers increases the difficulty for the platform to accurately estimate demand, and at the same time, causes the service fee to deviate from its optimal value, resulting in revenue loss. Pessimistically, Amin et al. [2] demonstrated that when the buyer possesses patience comparable to that of the seller, no learning algorithm can achieve sub-linear regret (cf. Appendix F.6).

In this paper, we address the above challenges by establishing the first set of theories for pricing service fees on third-party platforms. We summarize our contributions in the following.

- To the best of our knowledge, we are the first to attempt applying non-i.i.d. actions as instrumental variables in the problem of online pricing. In the traditional econometric framework [67], researchers often seek external instrumental variables to estimate the demand curve. However, this method has significant limitations because good instrumental variables are hard to find. We demonstrate that even when actions are not independent and identically distributed (i.i.d.) random variables, or even possess strong correlations with

one another, they still lead to excellent estimate of the demand curve. In Theorems 3.1, 3.2 and 4.3, we show our algorithms' optimal regret bounds.

- We discover that the randomness in supply can effectively assist us in learning the demand curve. This counterintuitive fact reveals why, in Theorem 3.2, we can achieve a regret of $\widetilde{\mathcal{O}}(1)$, but in the case where there is no noise in supply, in Theorem 3.1, we can only expect a regret of $\widetilde{\mathcal{O}}(\sqrt{T})$. We explain in Section 4 via lower bound results (Theorem 4.1 and 4.2) that these orders of magnitude differences are fundamental and unavoidable. Additionally, we detailed the phase transition points of regret bounds regarding supply randomness.

- We investigate robust pricing in the absence of prior knowledge of the buyer's discount rate. Our `AaI` and `AAaI` algorithms don't require the input of a discount rate to initiate, but can instead universally motivate buyers whose time has value to nearly truthfully report the demand curve. Specifically, our algorithm is also applicable in scenarios where the discount rate varies. The robustness of our algorithm is benign both in theory and applications.

## 1.1 Related Work

This work is closely related to instruments in machine learning [4, 5, 12, 56, 73, 47, 24, 69, 23], pricing with strategic buyers [55, 2, 29, 37, 1, 38], and demand learning under uncertainty [14, 26, 46, 21, 7, 48, 20, 50, 28]. Due to space constraints, additional references are provided in Appendix A for readers' reference. Detailed discussion of the relation and comparison between our work and previous work is also presented in Appendix A.

**Notations.** For any positive integer $n$, we let $[n]$ denote the set $\{1, ..., n\}$ and $n_1 : n_2$ represent $\{n_1, ..., n_2 - 1\}$. We use $\mathcal{N}(\mu, \sigma^2)$ to represent a Gaussian random variable with mean $\mu$ and variance $\sigma^2$. Moreover, we use $\mathcal{O}(\cdot)$ when ignoring constant terms while $\widetilde{\mathcal{O}}(\cdot)$ when ignoring constants and logarithmic terms. Similarly, we have $\Omega(\cdot), \widetilde{\Omega}(\cdot), \Theta(\cdot)$ and $\widetilde{\Theta}(\cdot)$.

## 2 Model and Assumptions

We consider an online dynamic pricing problem faced by a platform interacting with a representative buyer for $T$ rounds. The platform aims to maximize its revenue against the rational and strategic buyer by dynamically adjusting the service fees.

**Information structure of platform pricing.** In each round $t$, multiple sellers pose different portfolios on the platform and thus form the cumulative supply curve, denoted by $P_{St} = P_{St}(Q)$ as a function of $Q$. For example, Uber can observe in real-time how many different types of drivers are available on the platform, such as UberX, UberXL, Black SUV, etc. Similarly, Ticketmaster knows how many tickets are still available in each area. Then, the platform sets a service fee $a_t \in \mathbb{R}_{\geq 0}$. Here we assume the sellers are not strategic: as the platform formulates the fee-charging plan after observing $P_{St}(Q)$, there is no issue concerning trustfulness for sellers. In each round $t$, the representative buyer may receive a private signal, such as a shock on income, and form a time-dependent demand curve, denoted by $P_{Dt} = P_{Dt}(Q)$. However, the buyer may behave as if her demand is $P'_{Dt}$ rather than $P_{Dt}$ because of her forward-looking strategic behavior, elaborating below. Together with service fee $a_t$, the platform observes an equilibrium price and equilibrium quantity in the market, denoted by $(P^e_t, Q^e_t)$, satisfying

$$P^e_t := P_{St}(Q^e_t) + a_t = P'_{Dt}(Q^e_t). \tag{1}$$

Then, the platform receives revenue $\Pi_t = a_t \cdot Q^e_t(P_{St}, P'_{Dt}, a_t)$ in this round. As a result, across the time horizon of $T$, the platform has a cumulative revenue: $\sum_{t=1}^{T} a_t \cdot Q^e_t(P_{St}, P'_{Dt}, a_t)$. As a concrete setting, we study the case when both the supply and demand curves have linear forms following canonical literature of both demand learning [19, 61, 44, 74] and instrumental variable models [56].

**Assumption 2.1.** Assume that the supply curve in round $t$ has the form of

$$P_{St}(Q) = \alpha_0 + \alpha_1 Q + \epsilon_{St}$$

and the demand curve is

$$P_{Dt}(Q) = \beta_0 + \beta_1 Q + \epsilon_{Dt}.$$

The independent noise terms $\epsilon_{St}$ and $\epsilon_{Dt}$ follow normal distributions, $\mathcal{N}(0, \sigma_S^2)$ and $\mathcal{N}(0, \sigma_D^2)$, respectively.

To avoid nuisance from market feasibility, we assume that $\beta_0 > \alpha_0 \geq 0$, $\alpha_1 \geq 0 > \beta_1$, and that all equilibrium prices and quantities are positive and bounded. We note from Equation (1) that if $Q_t^e$ is negative, then optimal $Q_t^e$ shall be truncated as 0. For simplicity of analysis, we will not make truncation throughout the paper. This shall not influence the validity of our analysis and results.

**Utility-maximizing buyer.** We assume the representative buyer fully knows the learning policy used by the platform to set service fees [37]. Note that the buyer is only aware of the policy beforehand. If the policy involves randomization, the buyer knows in advance the policy but not the realization of the policy. In fact, previous behavior-based pricing literature [41, 64, 10, 11] has suggested that committing to a pricing strategy can help the platform earn more revenue.

In each time period $t$, the buyer receives a surplus, say $\text{surplus}_t(P_{St}, P_{Dt}, P_t^e, Q_t^e, a_t)$, which depends on the supply curve, demand curve, equilibrium price, equilibrium quantity, and the service fee. For brevity, we will write $\text{Sur}_t$ as an abbreviation of $\text{surplus}_t(P_{St}, P_{Dt}, P_t^e, Q_t^e, a_t)$. We utilize the Ramsey model [62], which originates from the economic literature, to calculate the surplus. More rigorously, we employ Assumption 2.2. We postpone the discussion of the economic intuition behind Assumption 2.2 and its implications for calculating the surplus to Appendix B for interested readers.

**Assumption 2.2.** We assume a private market where all sellers are civilian-run enterprises.

The representative buyer has a discount rate $\gamma \in [0, 1)$. When $\gamma = 0$, the buyer is myopic and only considers her surplus in the current round $t$. She truthfully purchases the optimal quantity of items she needs. That is, her realized demand curve $P'_{Dt}$ coincides with $P_{Dt}$. However, when $\gamma \in (0, 1)$, i.e., the buyer is far-sighted [8, 72], she may choose to *misreport* her demand curve $P'_{Dt} \neq P_{Dt}$ in exchange for increasing long-term expected cumulative utility: $\mathbb{E}[\sum_{s=t}^T \gamma^{s-t} \text{Sur}_s]$, where again $\text{Sur}_s$ is the surplus gained at time $s$. The expectation is taken over the randomness for all time $s > t$: supply randomness $\epsilon_{Ss}$, demand randomness $\epsilon_{Ds}$, and potential randomness from the platform pricing policy. Here, we use the general economic term "misreport" to represent that the equilibrium price and quantity are not aligned with the buyer's true demand. They can be either larger or smaller. Note that the buyer won't show her demand curve $P_{Dt}(Q)$ to the platform at all, and all the information the platform can learn is through observing equilibrium prices and quantities. Thus, it is important for the platform to design a careful service charging mechanism to motivate a truthful disclosure of the demand curve from the buyer.

**Performance metric.** With the help of the revelation principle [63], there is an incentive-compatible-direct mechanism for pricing to achieve the highest revenue. So, we can define optimal service fee from Equation (1) by

$$a_t^* = \underset{a_t \geq 0}{\operatorname{argmax}} \, \mathbb{E}[a_t \cdot Q_t^e(P_{St}, P_{Dt}, a_t)].$$

The expectation is taken over $P_{Dt}$ since the platform needs to set $a_t$ before the realization of $P_{Dt}$. Define the suboptimality at time $t$ as

$$\text{SubOpt}_t(a_t) = \mathbb{E}[a_t^* \cdot Q_t^e(P_{St}, P_{Dt}, a_t^*) - a_t \cdot Q_t^e(P_{St}, P'_{Dt}, a_t)].$$

Our evaluation metric is then the revenue regret against a clairvoyant policy attained over the whole $T$ rounds, namely,

$$\text{Regret}(T) = \sum_{t=1}^T \text{SubOpt}_t(a_t).$$

**Remarks on the model.** We would like to provide some further remarks on the model as follows.

1. *Platform is patient while buyer's discount rate is $\gamma \in [0, 1)$.* In practice, platforms are usually less time-sensitive compared with individuals, for instance, in a sponsored search auction, where the platform usually auctions off large numbers of ad slots each time while buyers usually urgently need advertisement and value future rewards less. On the other hand, the platform is not especially concerned with slight decreases in immediate rewards and maybe more on user stickiness. Readers can refer to Drutsa [30], Golrezaei et al. [37] for more information on different time values for different market forces. Additionally, from a theoretical standpoint, achieving sub-linear regret is impossible when the buyer discount rate is one [2]. Therefore, our paper focuses on strategic buyers whose discount rates fall within the $[0, 1)$ range, accommodating both real-world scenarios and theoretical constraints.

2. *On and beyond the linear model.* In this paper, we focus on linear models. In practical scenarios, particularly when considering a localized segment of the market to avoid eliciting competitive reactions, there often emerges a pattern of linearity (which also serves as the cornerstone for the application of the so-called "Delta method" [45]). Further, Besbes and Zeevi [15] also suggests that in certain circumstances, misspecification stemmed from assuming a linear model is less detrimental than anticipated. We note that although we explicitly assume the linearity of the demand curve and the supply curve in the model, it should not be difficult to extend to some variations of non-linear models (e.g. log-log, semi-log model in hedonic pricing [66] and logistic model in click prediction [51]) like Ban and Keskin [13] with some extra assumption on non-linear factors, by utilizing the generalized method of moments (GMM). Our technique also has the potential to work in high-dimensional cases and even non-parametric Reproducing Kernel Hilbert Space (RKHS). Despite such extensions, we will adopt the linear model throughout this work for brevity and clarity. We leave further generalizations for future work.

## 3 Regret Upper Bounds: Actions as Instruments

In this section, we present our main algorithm $\texttt{AaI}$ (Action-as-Instruments) and demonstrate its theoretical guarantees. We assume that $T$ and $\sigma_S \geq 0$ are both known ($\sigma_D$ need not be known a priori). This assumption will be further relaxed in the next section.

---

**Algorithm 1** AaI Algorithm.

---
**Input:** $T$.
**Initialization:** $\mathcal{D} \leftarrow \emptyset$, $\mathcal{A} \leftarrow \emptyset$ and $t \leftarrow 0$.
**for** episode $m = 1, 2, \cdots$ **do**
    Estimate unknown parameters: $(\widehat{\beta}_0, \widehat{\beta}_1) \leftarrow \texttt{IV}(\mathcal{D}, \mathcal{A})$ (Algorithm 4 in Appendix C.1).
    Form estimate for the demand curve $\widehat{P}_D = \widehat{\beta}_0 + \widehat{\beta}_1 Q$.
    Initialize datasets $\mathcal{D} \leftarrow \emptyset$ and $\mathcal{A} \leftarrow \emptyset$.
    **for** $\tau = t + 1$ **to** $t + 2^m$ **do**
        Observe the supply curve $P_{St} = \alpha_t + \alpha_1 Q$.
        Select the action $a_t \leftarrow \texttt{Act}(P_{St}, \widehat{P}_D)$ using Algorithm 2.
        Observe equilibrium $o_t = (P_t^e, Q_t^e)$.
        Update $\mathcal{D} \leftarrow \mathcal{D} \cup o_t$ and $\mathcal{A} \leftarrow \mathcal{A} \cup a_t$.
    **end for**
    Update round index: $t \leftarrow t + 2^m$.
**end for**

---

**Algorithm 2** Act Algorithm.

---
**Input:** $P_S$ and $P_D$.
Calculate $\widehat{a}^* \leftarrow \arg\max_{a \geq 0} \; a \cdot \widehat{Q}^e(a)$, where $\widehat{Q}^e(a)$ is decided by $P_S(\widehat{Q}^e) + a = \widehat{P}_D(\widehat{Q}^e)$.
Generate an independent noise term $\epsilon \sim \mathcal{N}(0, \frac{1}{\sqrt{\text{size}(\mathcal{A})+1}})$
Set service fee $a \leftarrow \widehat{a}^* + \epsilon \cdot \mathbb{1}\{\sigma_S = 0\}$.
**Output:** $a$.

---

We now elaborate on the design of $\texttt{AaI}$ and how it overcomes the main challenges mentioned in Section 1. We first demonstrate the design of our low-switching regime. In Algorithm 1, to deal with the far-sighted buyer, we update our policy when $t$ is a power of 2. The intuition behind the low-switching cost update is that since the buyer has a discount rate strictly lower than 1, long-run benefits will be hard to compensate for short-run loss. Under a low-switching regime, the gap between the buyer's behavior $P_D'$ and the true demand curve $P_D$ becomes small. In the existing literature, Ai et al. [1] utilizes explicit "buffer" periods against far-sighted buyers while other works such as Golrezaei et al. [37, 38] use implicit methods to motivate truthfulness. In Algorithm 1, we adopt an implicit way to punish untruthful behavior so that we don't need information about the buyer's discount rate, thus enhancing the robustness and universality of our algorithm. We note that the use of low-switching cost algorithms may not be necessary when facing a myopic buyer, though indispensable for a far-sighted buyer [34, 71].

Second, the proof of Theorem 3.1 and 3.2 relies on solving the challenge that the platform can only observe equilibrium prices and quantities. The novel approach we take here is to utilize the service fee price as an instrumental variable (reflected in Algorithm 4). Despite the fact that the service fees as actions are correlated, we prove they provide valid estimates of the true coefficients $(\beta_0, \beta_1)$, which is rigorously shown in Lemma 3.1 (in the lemma only $\beta_1$ is analyzed; a similar result also holds for $\beta_0$).

**Lemma 3.1.** *Let $\widehat{\beta}_{1t}$ be the estimated slope after round $t$. Then, under Assumptions 2.1 and 2.2, with high probability, it holds that*

$$|\widehat{\beta}_{1t} - \beta_1| \lesssim \mathcal{O}(\frac{\sqrt{\log\log T}}{t^{\frac{1}{4}}} + \frac{1}{(1-\gamma)\sqrt{t}}) \text{ for all } t \in [T] \text{ when } \sigma_S = 0,$$

*and*

$$|\widehat{\beta}_{1t} - \beta_1| \lesssim \mathcal{O}(\frac{\sqrt{\log\log T}}{\sigma_S\sqrt{t}} + \frac{1}{\sigma_S^2(1-\gamma)t}) \text{ for all } t \in [T] \text{ when } \sigma_S > 0.$$

Now let's discuss the magnitude of the artificially added noise (see Algorithm 2) — which serves as an exploration step to learn demand information. Combined with our previous discussion, we will present our main theorems in this section. We differentiate between two cases: $\sigma_S = 0$ and $\sigma_S > 0$.

## 3.1  $\sigma_S = 0$: $\widetilde{\mathcal{O}}(\sqrt{T})$ **Regret**

We first consider the case when there is no supply randomness. At the beginning of each episode $m$, we add a $\mathcal{O}(1)$ noise to the service fee. Then, we decay the magnitude of noise variance at an inverse square root rate. There are two key points when implementing the algorithm.

First, there is a trade-off between utilizing early data and recent data — the more data we use, the better estimate we will have, but the early data may have bad data-generating processes which may cause extra nuisances. Therefore, we choose only to use data from the last episode, whose length is roughly half of all available data and we reset the magnitude of noise as long as we start a new episode to ensure sufficient exploration.

Second, there is a trade-off between adding larger and smaller randomness — the more randomness we add to the service fee, the more accurate estimate we will obtain, whereas higher noise leads to larger revenue loss. In our design, in episode $m$, the variance level of noise added is designed to be $\mathcal{O}(\frac{1}{\sqrt{2^m}})$ on average where $2^m$ is the length of the episode. It decays at a relatively slow rate, aiming at exploration without losing much exploitation. We note that a special case of $\sigma_S = 0$ is when the supply curve coincides with the $Q$-axis ($P_{St}(Q) = 0$) and the buyer has no strategic behavior ($\gamma = 0$). This degenerates to the standard dynamic pricing problem [44] where a $\widetilde{\mathcal{O}}(\sqrt{T})$ regret holds. The following theorem shows that with the additional features in our model (equilibrium observation, strategic behavior), an $\widetilde{\mathcal{O}}(\sqrt{T})$ regret bound still holds.

**Theorem 3.1** ($\sigma_S = 0$). *Under Assumptions 2.1 and 2.2, for any fixed failure probability $\iota \in (0, 1)$, with probability at least $1 - \iota$, Algorithm 1 achieves at most $\mathcal{O}(\sqrt{T}\log(\frac{\log T}{\iota}) + \frac{\log T}{(1-\gamma)^2})$ regret against any buyer whose discount rate $\gamma \in [0, 1)$ when supply doesn't have noise, i.e. $\sigma_S^2 = 0$. Here, $\mathcal{O}(\cdot)$ hides only absolute constants.*

## 3.2  $\sigma_S > 0$: **Noise Helps Learning**

We then consider the case when there is supply randomness. In contrast to Section 3.1, here no random noise is injected into the empirical optimal action $\widehat{a}^*$ (see Algorithm 2). The following theorem shows that noise helps learning — an $\widetilde{\mathcal{O}}(1)$ regret is obtainable.

**Theorem 3.2** ($\sigma_S > 0$). *Under Assumptions 2.1 and 2.2, for any fixed failure probability $\iota \in (0, 1)$, with probability at least $1 - \iota$, Algorithm 1 achieves at most $\mathcal{O}(\frac{\log T\log(\frac{\log T}{\iota})}{\sigma_S^2} + \frac{\log T}{\sigma_S^2(1-\gamma)})$ regret against any buyer whose discount rate $\gamma \in [0, 1)$. Here, $\mathcal{O}(\cdot)$ hides only absolute constants.*

The regret upper bound in Theorem 3.2 contains two parts. The first $\mathcal{O}(\log T)$ term is the main regret incurred by learning the demand curve — which illustrates how noise helps learning. Although the

platform can only observe equilibrium prices and quantities, the extra randomness in the supply curve automatically generates exploration and pushes the empirical optimal action $\hat{a}^*$ from Algorithm 2 to vibrate around some intrinsic number. As a result, the estimation of $\beta$ generates a fast convergence rate of $\mathcal{O}(1/\sqrt{t})$ (see Lemma 3.1) even if no active exploration is presented. As a comparison, in Section 3.1 when the supply curve is deterministic, the problem becomes "degenerate" to some extent. The equilibrium prices on the supplier side and equilibrium quantities lie on one line (the fixed supply curve) in the 2-dimensional space, which forces us to do active exploration that leads to a $\mathcal{O}(\sqrt{T})$ regret.

The second term is the extra regret due to the strategic behavior of the buyer. Golrezaei et al. [37] achieves an extra $\mathcal{O}(\frac{\log^2 T}{\log^2(1/\gamma)})$ regret in repeated auction pricing when the discount rate is approaching one. Since $\log(1/\gamma) \le 2(1-\gamma)$ for $\gamma \in [0,1)$, we obtain better results in both the order of $\log T$ and order of $\frac{1}{1-\gamma}$. We note that since the implementation of Algorithm 1 doesn't depend on the buyer's discount rate $\gamma$, it can achieve $\mathcal{O}(\log T \log \log T)$ regret even if the discount rates are changing in different rounds — as long as the discount rate $\gamma_t$ has a uniform upper bound $\bar{\gamma}$ and $\bar{\gamma} < 1$, which is widely observed in the real-world market, especially online advertising [31], Algorithm 1 remains good permanence.

From Theorems 3.1 and 3.2, we know the key to achieve $\widetilde{\mathcal{O}}(1)$ regret is the internal randomness of the supply curve. Therefore, we have the following corollary.

**Corollary 3.2.** *If the platform needs to set a uniform service fee in multi-markets, such as Order Processing Fee on Ticketmaster [17], the demand curve goes to*

$$\vec{P}_{Dt} = \vec{\beta}_0 + \vec{\beta}_1 \otimes \vec{Q} + \vec{\epsilon}_{Dt},$$

*where $\otimes$ is the Kronecker product of two matrices. Then, as long as in one market, the corresponding supply has internal randomness, Algorithm 1 can achieve $\widetilde{\mathcal{O}}(1)$ regret by utilizing $a_t$ as an instrument for all markets. Otherwise, it will suffer $\widetilde{\mathcal{O}}(\sqrt{T})$ regret.*

## 4 Regret Lower Bounds: Phase Transition

In this section, we present results on regret lower bounds. We will focus on the case when $\gamma = 0$ and investigate how the regret intrinsically scales as a function of $T$ as well as $\sigma_S$. We will show that there exhibits a phase transition phenomenon with respect to $T$ and an inverse square law with respect to $\sigma_S$. We first present a result when there is no supply randomness.

**Theorem 4.1.** *Given any time horizon $T$, when the supply doesn't have noise, the worst-case expected regret is lower bounded by $\Omega(\sqrt{T})$. In $\Omega(\cdot)$ we are hiding a constant term irrelevant with $\sigma_S$ and $T$.*

Next, we provide a lower bound when $\sigma_S$ can take any general positive number.

**Theorem 4.2.** *Given any time horizon $T$ and the supply noise level $\sigma_S$, the worst-case expected regret is lower bounded by*

$$\Omega\left(\sqrt{T} \wedge \frac{\log T}{\sigma_S^2(1 + \log_+(1/\sigma_S))}\right),$$

*where $\log_+(\cdot) = \max\{0, \log(\cdot)\}$. In $\Omega(\cdot)$ we are hiding a constant term irrelevant with $\sigma_S$ and $T$.*

The proof of Theorem 4.1 depends on constructing two hard-to-differentiate instances such that the demand curves deviate with each other while the supply curves remain the same. The proof of Theorem 4.2 relies on the multivariate Van Trees inequality [36, 44]. An important step in both of the proof is when considering the magnitude of demand noise $\sigma_D$, it should be dependent on $\alpha_1$ and $\beta_1$. The reason is that if the noise magnitude is irrelevant with the slope parameters, the standard deviation of equilibrium prices and quantities observed by the platform can provide additional information on the true parameters — this will make it possible for a policy to learn more quickly by "cheating". Only when the noise magnitude posits a delicate dependence on the true parameters, the equilibrium prices/quantities exhibit constant deviation across all instances without information leakage.

Now we give some remarks for the theorems. Theorem 4.1 states that the regret order in Theorem 3.1 is tight. It also implicitly provides some intuition for our choice of the magnitude of the artificially

introduced noise in our algorithm design. The total variance of noise we add is proportional to $\sqrt{T}$ — this shall be the largest noise magnitude we shall use for sufficient exploration to match the $\Omega(\sqrt{T})$ lower bound.

Theorem 4.2 shows that if $\sigma_S > 0$ is a constant irrelevant with $T$, then our regret upper bound in Theorem 3.2 is tight up to a $\log T$ factor. Moreover, Theorem 4.2 gives a valid lower bound if $\sigma_S$ is entangled with $T$. To be precise, if $\sigma_S^2 \lesssim \mathcal{O}(1/\sqrt{T})$, then the best we can hope is an $\widetilde{\mathcal{O}}(\sqrt{T})$ regret. Only when $\sigma_S^2 \gtrsim \omega(1/\sqrt{T})$ can we expect a regret bound better than $\widetilde{\mathcal{O}}(\sqrt{T})$.

Readers may immediately notice that there is a gap between our regret upper bounds (see Theorem 3.2) and the regret lower bound (see Theorem 4.2) when $\sigma_S^2 \lesssim \mathcal{O}(1/\sqrt{T})$. It raises the following question whether we can achieve the same regret upper bounds without knowing $\sigma_S$ in advance. The answer is YES! We have the following theorem.

**Theorem 4.3.** *Under Assumptions 2.1 and 2.2, there exists an algorithm, e.g., Algorithm 3, whose expected regret is at most $\widetilde{\mathcal{O}}(\sqrt{T} \wedge \sigma_S^{-2})$ against any buyer with discount rate $\gamma \in [0,1)$. Here, $\widetilde{\mathcal{O}}(\cdot)$ hides only constants and logarithmic terms.*

---

**Algorithm 3** AAaI (Adaptive Action-as-Instruments) Algorithm.

---

**Input:** $T$.
**for** $t = 1$ to $T_0 \asymp \Theta(\log T)$ **do**
    Observe market randomness $\epsilon_{St}$ **or** supply intercept $\alpha_0 + \epsilon_{St}$.
**end for**
**Hypothesis Test:** $\mathbb{H}_0 : \sigma_S^2 \lesssim \mathcal{O}(\frac{1}{\sqrt{T}})$ and $\mathbb{H}_1 : \sigma_S^2 \gtrsim \Omega(\frac{1}{\sqrt{T}})$, denoting the result as $\mathbb{H}$.
Conduct $\texttt{AaI}(T - T_0)$ with $\texttt{Act-HT}(\mathbb{H}, \cdot, \cdot)$ (Algorithm 5 in Appendix C.2) replacing $\texttt{Act}(\cdot, \cdot)$.

---

In Algorithm 3, $\texttt{Act-HT}$ is essentially a generalized version of $\texttt{Act}$: if $\mathbb{H}_0$ holds, we treat $\sigma_S$ as if it is 0; if not, we treat $\sigma_S$ as it is. When $T$ is unknown or infinite, we can leverage the well-known doubling trick [9, 16] to achieve the same order of regret. From Theorem 4.3, we know that when the supply randomness is small enough, namely, $\sigma_S^2 \lesssim \mathcal{O}(1/\sqrt{T})$, the expected regret has a fixed rate $\widetilde{\Theta}(\sqrt{T})$, whereas when the supply randomness is large enough, namely $\sigma_S^2 \gtrsim \Omega(1/\sqrt{T})$, the expected regret is inversely proportional with respect to $\sigma_S^2$ — an inverse square law ignoring constants and logarithmic terms. Therefore, there exists an essential phase transition when $\sigma_S^2 \asymp \Theta(1/\sqrt{T})$ (see Figure 4) — this tells us that we should take market randomness into consideration whenever solving pricing problems.

## 5 Numerical Study

In this section, we conduct two simulation experiments. The goal is to test the performance of our algorithms as well as numerically demonstrate the phase transition phenomenon. We provide detailed experimental descriptions in Appendix G.

In the first experiment, we consider regret attained in Algorithm 1 under two scenarios. Here, we set $T = 10^5$. We replicate 10 times in each setting and draw the average regrets and their 95% confidence regions. We consider both constant $\sigma_S^2$ and zero randomness in Figure 2. It's obvious that when $\sigma_S^2 \asymp \Theta(1)$, the growth rate of regret is significantly smaller than the growth rate when $\sigma_S^2 = 0$. The first regret is slightly larger than 200 and the second one is more than 1200, validating the respective $\mathcal{O}(\log^2 T)$ and $\mathcal{O}(\sqrt{T} \log T)$ expected regrets. Numerically, when $\sigma_S^2 = 1$, we find that the regrets when $T = 20000, 40000, 60000, 80000, 100000$ ($\log T = 9.90, 10.60, 11.00, 11.29, 11.51$) are 220, 230, 234, 236, and 237, respectively. These points are even slightly sublinear. So, the actual performance is even better than the theoretical bound. In addition, when there is no randomness in the supply, i.e., $\sigma_S^2 = 0$, we have $\log(\text{Regret}) = 6.43, 6.75, 6.95, 7.10, 7.17$, respectively. The estimated slope by linear regression is 0.47, testifying our regret bound.

Moreover, we increase the number of trajectories to 100 and observe that the bandwidth of the corresponding confidence region significantly decreases (cf. Figure 3 (top, right)). Additionally, we test the regret under different $\sigma_S^2$ values, say 0.5, 1, 1.5 and 2. We notice that the larger the $\sigma_S^2$, the smaller the regret, which confirms our theoretical results (cf. Figure 3).

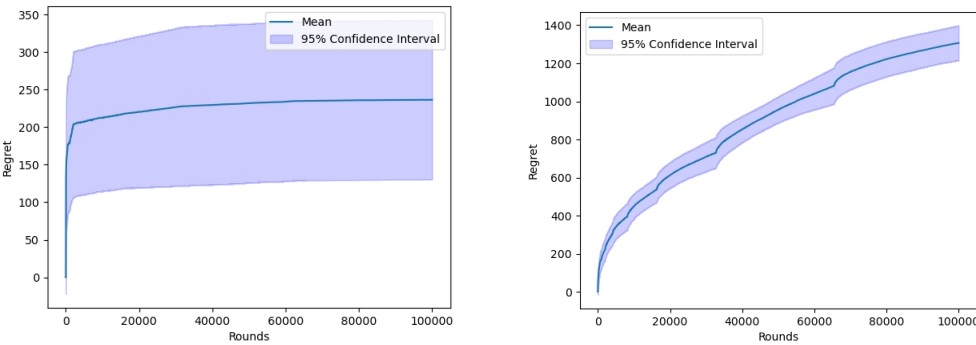

Figure 2: 95% confidence region of regret of Algorithm 1 over 10 trajectories: $\sigma_S^2 = 1$ (left) and $\sigma_S^2 = 0$ (right).

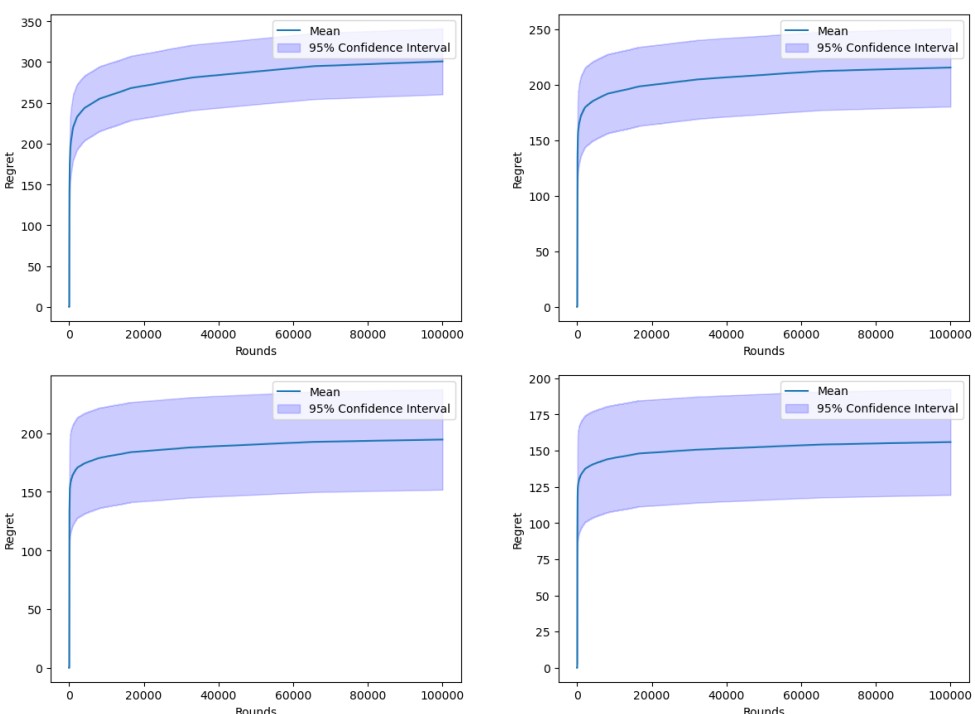

Figure 3: 95% confidence region of regret of Algorithm 1 over 100 trajectories: $\sigma_S^2 = 0.5, 1, 1.5, 2$ (top to bottom, left to right).

Then, we conduct the second experiment examining the dependency of regret on supply randomness $\sigma_S^2$. We set $T = 10^4$ and range $\sigma_S^2$ from 0.001 to 1. We replicate the simulation 20 times and use quantile statistics to enhance the robustness (see raw results in Figure 5, i.e., blue points). There are around 100 points choosing $\mathbb{H}_0$ and we can observe the phase transition when $\sigma_S^2 \approx 0.1$. Notice that Algorithm 3 has a $\mathcal{O}(\sqrt{T} \log T)$ expected regret for $\mathbb{H}_0$ but $\mathcal{O}(\sqrt{T} \log^2 T)$ for $\mathbb{H}_1$ when $\sigma_S^2 \asymp \Theta(\frac{1}{\sqrt{T}})$ (cf. Appendix F.5). This additional $\log T$ factor explains why there is increasing fluctuation in regret near the phase transition point, as Algorithm 3 engages in a mixture of applying $\mathbb{H}_0$ and $\mathbb{H}_1$. Finally, we employ locally weighted scatterplot smoothing (LOWESS) [25] to approximate regret for each level of randomness $\sigma_S^2$, with the fitting depicted by a red line. We notice that regret first reaches a plateau and keeps nearly constant and then gradually decreases, testifying Theorem 4.3.

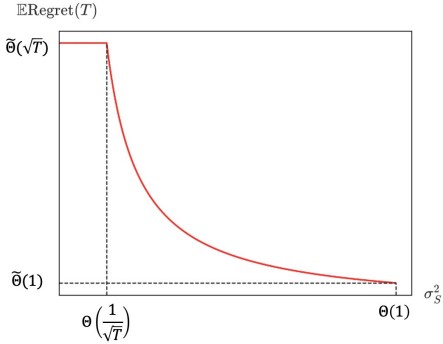
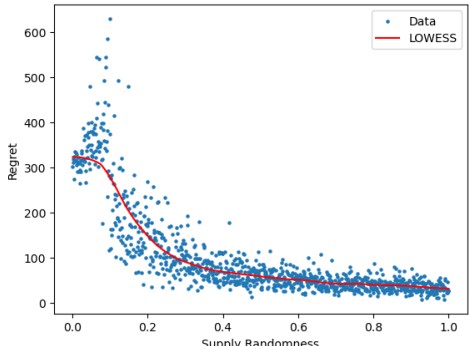

Figure 4: Phase transition with supply randomness $\sigma_S^2$. Aggregating Theorems 4.1 to 4.3.

Figure 5: Phase transition in Algorithm 3 and its non-parametric local fitting.

# 6   Conclusion

In this paper, we study the dynamic pricing problem for third-party platforms. Our model incorporates practical challenges involving lack of demand knowledge, limited observation of equilibria, and buyer strategic behavior. We design an effective policy that obtains optimal performance guarantees to address the challenges by injecting carefully chosen randomness, using non-i.i.d. actions as instruments, and forcing a low-switching design. Specifically, in the case of supply fluctuations, we achieve a regret upper bound of $\widetilde{\mathcal{O}}(1)$, and when supply is fixed, we achieve a regret bound of $\widetilde{\mathcal{O}}(\sqrt{T})$, both of which match the information-theoretical lower bounds. Additionally, we demonstrate the relationship between regret and supply randomness, and provide their optimal dependency and phase transition points.

Questions arise for future explorations. What is the dependence of regret on the discount rate $\gamma$? Our conjecture is that a discount rate strictly less than 1 will introduce inevitable $\Omega(\frac{1}{1-\gamma})$ regret universally, meaning the regret upper bound in Theorem 3.2 is optimal with respect to $\gamma$ but the bound in Theorem 3.1 may not. Unfortunately, the analysis can be very challenging, which we leave as an open question. Furthermore, one future work is to extend the linear model to more complex models and investigate whether the insights in this paper (e.g., phase transition) still hold. Despite the simplicity of our model, we hope our results offer valuable insights into solving general service fee pricing problems for third-party platforms.

# Acknowledgments

The authors would like to thank the reviewers for their valuable comments and suggestions, which have greatly improved the article.

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

# A Literature Review

This work contributes to the theory of econometrics, demand learning, and pricing with strategic buyers. We summarize below three lines of existing literature pertinent to our work.

**Instruments in Machine Learning.** Instrumental variable is a powerful tool in econometrics [4, 5, 12]. Nambiar et al. [56] uses IV to learn the demand curve under model misspecification, inspired by Petrin and Train [59], Phillips et al. [60]. In recent years, a series of articles have emerged that use the IV method to estimate machine learning parameters. Yu et al. [73], Liao et al. [47] utilize instruments to learn a nearly-optimal policy via offline data. Chen and Qi [24], Uehara et al. [69] adopt a non-parametric instrumental variable framework to do off-policy evaluation. Chen and Zhang [23] considers a time-varying instrument and Singh [65] extends the kernel methods to scenarios with confounders.

Unlike existing literature, our paper first leverages endogenous proactive actions as IV in online learning problems, especially in pricing. Unlike offline data, the distribution of our instrument variable is not only non-identical but also even non-independent. Meanwhile, as we face a revenue-maximizing problem, we need to deal with the famous exploration-exploitation tradeoff. More efficient instruments may decrease the estimation error leading to higher future revenue, but usually suffer a larger short-term loss. Thus, prior research does not encompass our model and we examine the robustness of IV to market randomness with novelty.

**Pricing with Strategic Buyers.** There is a burgeoning amount of literature on pricing with strategic buyers. Amin et al. [2] first proves that no algorithm can achieve sublinear regret when buyers are as patient as sellers. Deng et al. [29] considers pricing in dynamic mechanism design with less patient buyers. It obtains sublinear regret in contextual auctions. Golrezaei et al. [37] study optimal reserve design problem original from Myerson [55] facing strategic bidders while Mohri and Munoz [54] considers a corresponding revenue optimization problem. Golrezaei et al. [38] considers pricing with strategic buyers under non-parametric market noise and Ai et al. [1] extends to Markov decision process (MDP) pricing models. Mohri and Medina [53], Kanoria and Nazerzadeh [43], Epasto et al. [32], Amin et al. [3] also study such issues under different information structures and depict different scenarios in real markets.

The difficulty in our model is that we can only observe equilibrium prices and quantities. Worst yet, there are confounders behind the information feedback. Therefore, the methods from previous literature cannot be directly applied to our model. Consequently, we explore a robust pricing framework originating from econometrics to address this issue.

**Demand Learning under Uncertainty.** Demand learning is a hot topic in microeconomics, management science, and operations research [14, 26, 46, 21, 7, 48, 20]. Lobo and Boyd [50] considers a linear demand model and invents a "price-dithering" policy to add perturbation. Den Boer and Zwart [28] invents the controlled variance pricing idea and Broder and Rusmevichientong [18] scales exploration by adding $t^{-1/4}$ which has similar connotations to our approach.

Nambiar et al. [56] considers confounders in demand learning but doesn't involve strategic buyers. We extend high-level econometrics ideas behind it and design recipes against non-truthful buyers. We construct a robust pricing framework free of prior knowledge of both discount info and market randomness and attain optimal regret bounds across all settings.

# B Discussion on Assumption 2.2

We clarify the definition of $\text{Sur}_t$ first. Similar to the Ramsey model [62], there are three different types of sellers, namely, civilian-run enterprises, state-owned enterprises, and mixed-ownership enterprises [57]. For civilian-run enterprises, 100% of the profits belong to individuals, namely the representative buyer, so she aims to maximize the sum of producer surplus and consumer surplus. Then, it holds that

$$\text{Sur}_t = \text{Consumer surplus at time } t + \text{Producer surplus at time } t.$$

Meanwhile, for state-owned enterprises, their goal is to maximize their own scale. This is a common form of business organization, especially in developing countries, such as China (see Modigliani and Cao [52] for more information). Because the buyer cannot receive dividends from companies, she

will only maximize their consumer surplus, namely,

$$\text{Sur}_t = \text{Consumer surplus at time } t.$$

Finally, for mixed-ownership enterprises, a portion of the profits will be distributed to the buyer in the form of dividends. We use $\alpha \in (0,1)$ to denote the proportion. Therefore, we have

$$\text{Sur}_t = \text{Consumer surplus at time } t + \alpha * \text{Producer surplus at time } t.$$

Hence, we make Assumption 2.2 in our paper, all sellers are civilian-run enterprises. We point out that this assumption won't affect our results and regrets with the same order can be obtained under state-owned enterprises and mixed-ownership enterprises. Here, we use the definition of $\text{Sur}_t$ to detail Assumption 2.2 as Assumption B.1.

**Assumption B.1.** Without loss of generality, we assume that all sellers are civilian-run enterprises. Then, the strategic representative buyer aims to maximize her cumulative surplus aligned with the sum of the corresponding consumer surplus and provider surplus. In other words, $\text{Sur}_t$ has the form of

$$\text{Sur}_t = \text{Consumer surplus at time } t + \text{Producer surplus at time } t.$$

## C  Omitted Details of Algorithm Implementations

### C.1  Omitted Details of Algorithm 1

In this section, we present the subroutine used in Algorithm 1. Specifically, Algorithm 4 details the coefficient estimation process.

---

**Algorithm 4** IV Algorithm.

---

**Input:** $\mathcal{D}$ and $\mathcal{A}$.
Regress $\{Q^e, P^e\}$ on $\{a_t\}$: $Q^e = \widehat{a}_0 + \widehat{a}_1 a$, $P^e = \widehat{b}_0 + \widehat{b}_1 a$.
Estimate $\beta_1$: $\widehat{\beta}_1 = \frac{\widehat{b}_1}{\widehat{a}_1}$.
Calculate sample means[1]: $\bar{Q}^e = \frac{1}{\text{size}(\mathcal{A})} \sum_t Q^e$ and $\bar{P}^e = \frac{1}{\text{size}(\mathcal{A})} \sum_t P^e$.
Estimate $\beta_0$: $\widehat{\beta}_0 = \bar{P}^e - \widehat{\beta}_1 \bar{Q}^e$.
**Output:** $(\widehat{\beta}_0, \widehat{\beta}_1)$.

---

### C.2  Omitted Details of Algorithm 3

We fill in the details of Algorithm 3 below.

---

**Algorithm 5** Act-HT Algorithm.

---

**Input:** $\mathbb{H}$, $P_S$ and $P_D$.
Calculate $\widehat{a}^* \leftarrow \text{argmax}_{a \geq 0} \ a \cdot \widehat{Q}^e(a)$, where $\widehat{Q}^e(a)$ is decided by $P_S(\widehat{Q}^e) + a = \widehat{P}_D(\widehat{Q}^e)$.
Generate an independent noise term $\epsilon \sim \mathcal{N}(0, \frac{1}{\sqrt{\text{size}(\mathcal{A})+1}})$
Set service fee $a \leftarrow \widehat{a}^* + \epsilon \cdot \mathbb{1}\{\mathbb{H} = \mathbb{H}_0\}$.
**Output:** $a$.

---

## D  Concentration Inequalities

We now recall some inequalities [70] which bound the accumulated difference of a martingale and are extensively used in our proofs.

---

[1]We define $\frac{0}{0} = 0$ for convenience.

**Lemma D.1** (Bernstein-type bound for a martingale difference sequence). *Let $\{(D_k, \mathcal{F}_k)\}_{k=1}^{\infty}$ be a martingale difference sequence, and suppose that $\mathbb{E}\left[e^{\lambda D_k} \mid \mathcal{F}_{k-1}\right] \leq e^{\lambda^2 v_k^2/2}$ almost surely for any $|\lambda| < 1/\alpha_k$. Then the following hold:*

*(a) The sum $\sum_{k=1}^{n} D_k$ is sub-exponential with parameter tuple $\left(\sqrt{\sum_{k=1}^{n} v_k^2}, \alpha_*\right)$, where $\alpha_* := \max_{k=1,\ldots,n} \alpha_k$.*

*(b) The sum satisfies the concentration inequality*

$$\mathbb{P}\left[\left|\sum_{k=1}^{n} D_k\right| \geq t\right] \leq \begin{cases} 2e^{-\frac{t^2}{2\sum_{k=1}^{n} v_k^2}} & \text{if } 0 \leq t \leq \frac{\sum_{k=1}^{n} v_k^2}{\alpha_*} \\ 2e^{-\frac{t}{2\alpha_*}} & \text{if } t > \frac{\sum_{k=1}^{n} v_k^2}{\alpha_*} \end{cases}$$

**Lemma D.2** (Hoeffding bound). *Suppose that the variables $X_i, i = 1, \ldots, n$, are independent, and $X_i$ has mean $\mu_i$ and sub-Gaussian parameter $\sigma_i$. Then for all $t \geq 0$, we have*

$$\mathbb{P}\left[\left|\sum_{i=1}^{n}(X_i - \mu_i)\right| \geq t\right] \leq 2\exp\left\{-\frac{t^2}{2\sum_{i=1}^{n}\sigma_i^2}\right\}.$$

# E    Omitted Proof in Section 3

We first prove results in Section 3.2 and then prove results in Section 3.1 in leverage of some lemmas derived from the proof of Theorem 3.2.

## E.1    Omitted Proof in Section 3.2

Since we assume that all equilibrium prices and quantities are bounded, reflecting the buyer's budget constraint and sellers' production capacity, we use $\bar{P}$ and $\bar{Q}$ to represent their upper bounds respectively. Moreover, recall that the buyer is myopic, so the demand curve behaved $P_D'$ is the same as the real demand $P_D$.

We first consider the situation in which the buyer is myopic as a warm-up. We have the following theorem.

**Theorem E.1.** *Under Assumption 2.1, for any fixed failure probability $\iota \in (0,1)$, with probability at least $1 - \iota$, Algorithm 1 achieves at most $\mathcal{O}(\frac{\log T \log(\frac{\log T}{\iota})}{\sigma_S^2})$ regret, where $\mathcal{O}(\cdot)$ hides only absolute constants when facing a myopic buyer.*

### E.1.1    Useful Facts for Proving Theorem E.1

Now, we state the following lemma for the planning problem first.

**Lemma E.1.** *If $\beta_0$ and $\beta_1$ are common knowledge, the optimal service fee is $a^* = \frac{\beta_0 - \alpha_0 - \epsilon_S}{2}$. Meanwhile, the corresponding equilibrium quantity is $Q^e = \frac{\beta_0 - \alpha_0 - \epsilon_S + 2\epsilon_D}{2(\alpha_1 - \beta_1)}$.*

*Proof.* Since $P_S = \alpha_0 + \alpha_1 Q + \epsilon_S$ and $P_D = \beta_0 + \beta_1 Q_D + \epsilon_D$, applying $P_S + a = P_D$ leads to $Q^e = \frac{\beta_0 - \alpha_0 - a + \epsilon_D - \epsilon_S}{\alpha_1 - \beta_1}$. As the platform cannot observe $\epsilon_D$ when setting the service fee $a$, i.e. $\epsilon_D$ is realized ex post, the expected revenue associated with $a$ is $\mathbb{E}a * Q^e = \frac{(\beta_0 - \alpha_0 - a - \epsilon_S)a}{\alpha_1 - \beta_1}$. It's maximized when

$$a^* = \frac{\beta_0 - \alpha_0 - \epsilon_S}{2}.$$

With some calculation, we know that the equilibrium quantity is

$$Q^e = \frac{\beta_0 - \alpha_0 - \epsilon_S + 2\epsilon_D}{2(\alpha_1 - \beta_1)}.$$

The reason why $\epsilon_S$ appears explicitly is that the shock on the supple curve is ex-ante and observed by the platform. □

Thereafter, we'd like to bound the estimation of $\beta_0$ and $\beta_1$ in Algorithm 4. We have the following proposition. We ignore the flooring in $\lfloor \log_2 T \rfloor$ throughout this section to avoid notation clutter.

**Proposition E.2.** *Let $\widehat{\beta}_1$ be the estimated slope of Algorithm 4 after the $m$-th episode. Then, under Assumption 2.1, there exists a constant $C_1$ such that with probability at least $1 - 6\delta$, we have*

$$|\widehat{\beta}_1 - \beta_1| \leq \frac{C_1 \sqrt{\log \log T}}{\sigma_S \sqrt{2^m}} \text{ for all } m \in [\log_2 T],$$

*then*

$$|\widehat{\beta}_{1t} - \beta_1| \lesssim \mathcal{O}(\frac{\sqrt{\log \log T}}{\sigma_S \sqrt{t}}) \text{ for all } t \in [T],$$

*where subscript $t$ represents which round.*

*Proof.* After the $m$-th episode, we know that we have data sets $\mathcal{D}$ and $\mathcal{A}$ with $\text{Size}(\mathcal{A}) = 2^m := n$. Then, we need to prove that there exists a uniform constant $C_1$ such that $|\widehat{\beta}_1 - \beta_1| \leq \frac{C_1 \sqrt{\log \log T}}{\sigma_S \sqrt{n}}$.

Since during episode $m$, we choose the service fee as $\frac{\widehat{\beta}_0 - \alpha_0 - \epsilon_S}{2}$, where $\widehat{\beta}_0$ is the latest estimate of $\beta_0$, i.e. the estimate after $m - 1$-th episode. With a little abuse of notation, we omit the subscript of it with $\widehat{\beta}_0$ along the road of proof without confusion.

When $\text{Size}(\mathcal{A}) = n$, it means that there are $n$ samples in the data set, that is to say, a trajectory of length $n$. We use $\mathbb{E}_n$ to represent the sample mean and $\mathbb{E}$ to represent the population mean. With simple algebra, we know that

$$\widehat{\beta}_1 = \frac{\mathbb{E}_n[P^e(a - \mathbb{E}_n a)]}{\mathbb{E}_n[Q^e(a - \mathbb{E}_n a)]} = \frac{\frac{1}{n} \sum_{i=1}^n P_i^e(a_i - \mathbb{E}_n a)}{\frac{1}{n} \sum_{i=1}^n Q_i^e(a_i - \mathbb{E}_n a)}.$$

Since $P^e = \beta_0 + \beta_1 Q^e + \epsilon_D$ as $(P^e, Q^e)$ is on the demand curve, it holds that

$$\widehat{\beta}_1 = \beta_1 + \frac{\frac{1}{n} \sum_{i=1}^n \epsilon_{Di}(a_i - \mathbb{E}_n a)}{\frac{1}{n} \sum_{i=1}^n Q_i^e(a_i - \mathbb{E}_n a)}.$$

First, we bound the numerator $\frac{1}{n} \sum_{i=1}^n \epsilon_{Di}(a_i - \mathbb{E}_n a) = -\frac{1}{2n} \sum_{i=1}^n \epsilon_{Di} \epsilon_{Si} + \frac{1}{2} \mathbb{E}_n \epsilon_S \mathbb{E}_n \epsilon_D$. We use $\sum_i x_i$ to denote $\sum_i \epsilon_{Di} \epsilon_{Si}$ for simplicity.

For $\frac{1}{n} \sum_i x_i$, we leverage Lemma D.1 to give a Bernstein-type bound for the corresponding martingale difference sequence.

**Lemma E.3.** *There exists constants $n_1$ and $\nu$ such that with probability at least $1 - \delta$, it holds that for any $n \geq n_1$ where $n = 2^m$ and $m \in [\log_2 T]$,*

$$|\frac{1}{n} \sum_{i=1}^n x_i| \leq \sqrt{\frac{2\nu^2 \log(\frac{2 \log_2 T}{\delta})}{n}}.$$

*Proof.* In our case, we know that $a_i$ depends on not only previous $\epsilon_D$ and $\epsilon_S$, but also $\epsilon_{Si}$. However, it doesn't rely on $\epsilon_{Di}$. This conditional independence motivates us to use actions as nearly valid instruments. Similarly, $x_i$ has the same property. Therefore, by utilizing this independence, we know that

$$\mathbb{E}[x_i \mid x_1, ..., x_{i-1}] = \mathbb{E}[\epsilon_{Di}] * \mathbb{E}[\epsilon_{Si} \mid x_1, ..., x_{i-1}] = 0.$$

As a result, $\{x_i\}_{i=1}^n$ becomes a martingale difference sequence and we use $\mathcal{F}$ to denote associated filtration. Then, let's compute the moment-generating function (MGF).

$$\mathbb{E}[e^{\lambda x_i} \mid \mathcal{F}_{i-1}] = \int \exp(\lambda \epsilon_S \epsilon_D) \frac{1}{\sqrt{2\pi}\sigma_S} e^{-\frac{\epsilon_S^2}{2\sigma_S^2}} \frac{1}{\sqrt{2\pi}\sigma_D} e^{-\frac{\epsilon_D^2}{2\sigma_D^2}} d\epsilon_S d\epsilon_D$$

$$= \frac{1}{\sqrt{1 - \lambda^2 \sigma_S^2 \sigma_D^2}}.$$

as long as $|\lambda| \leq \frac{1}{\sigma_S \sigma_D}$. The first equation comes from the definition of $\epsilon_D$ and $\epsilon_S$ while the second equation comes from simple algebra.

Therefore, assuming $\alpha = 2\sigma_S \sigma_D$, we know that for any $|\lambda| \leq \frac{1}{\alpha}$,

$$\mathbb{E}[e^{\lambda x_i} \mid \mathcal{F}_{i-1}] \leq \frac{1}{\sqrt{1 - \lambda^2 \sigma_S^2 \sigma_D^2}} \leq e^{\lambda^2 \sigma_S^2 \sigma_D^2}.$$

Then, assuming $\nu = \sqrt{2\sigma_S^2 \sigma_D^2} = \sqrt{2}\sigma_S \sigma_D$, it holds that $x_i$ is sub-exponential with parameters $(\nu, \alpha)$.

With the help of Lemma D.1, we know that as long as $n \geq \frac{2\alpha^2 \log(\frac{2 \log_2 T}{\delta})}{\nu^2} := n_1$, with probability at least $1 - \frac{\delta}{\log_2 T}$, $|\frac{1}{n} \sum_{i=1}^n x_i| \leq \sqrt{\frac{2\nu^2 \log(\frac{2 \log_2 T}{\delta})}{n}}$. Since there are at most $\log_2 T$ kinds of different values for $n$, the total sum of probabilities of bad events is less than $\frac{\delta}{\log_2 T} * \log_2 T = \delta$, which ends the proof. $\square$

For $\mathbb{E}_n \epsilon_S$, we have the following lemma therein to bound it.

**Lemma E.4.** *With probability at least* $1 - \delta$, *it holds that for any* $n = 2^m$ *and* $m \in [\log_2 T]$ *that*

$$|\frac{1}{n} \sum_{i=1}^n \epsilon_{Si}| \leq \sqrt{\frac{2\sigma_S^2 \log(\frac{2 \log_2 T}{\delta})}{n}}.$$

*Proof.* For any $n = 2^m$ where $m \in [\log_2 T]$, we know that $\epsilon_{Si}$ is a sub-Gaussian variable with parameter $\sigma_S$ because it follows distribution $\mathcal{N}(0, \sigma_S^2)$.

By applying Lemma D.2, namely Hoeffding's inequality, it holds that with probability $1 - \frac{\delta}{\log_2 T}$

$$|\frac{1}{n} \sum_{i=1}^n \epsilon_{Si}| \leq \sqrt{\frac{2\sigma_S^2 \log(\frac{2 \log_2 T}{\delta})}{n}}.$$

Combining all nuisance leads to the probability of all inequalities holding, that is, $1 - \delta$, which finishes our proof. $\square$

Moreover, for $\mathbb{E}_n \epsilon_D$, we have a similar result whence giving an upper bound for it.

**Lemma E.5.** *With probability at least* $1 - \delta$, *it holds that for any* $n = 2^m$ *and* $m \in [\log_2 T]$ *that*

$$|\frac{1}{n} \sum_{i=1}^n \epsilon_{Di}| \leq \sqrt{\frac{2\sigma_D^2 \log(\frac{2 \log_2 T}{\delta})}{n}}.$$

*Proof.* The proof is the same as the one for Lemma E.4 with replacement of $\epsilon_{Si}$ by $\epsilon_{Di}$. $\square$

Second, let's give some concentration bounds for the denominator $\frac{1}{n} \sum_{i=1}^n Q_i^e(a_i - \mathbb{E}_n a)$.

By applying Lemma E.1, we have the following decomposition of the denominator that

$$\frac{1}{n} \sum_{i=1}^n Q_i^e(a_i - \mathbb{E}_n a)$$

$$= \frac{1}{2n(\alpha_1 - \beta_1)} \left[ \sum_i 2(a_i - \mathbb{E}_n a)\epsilon_{Di} - \sum_i (a_i - \mathbb{E}_n a)\epsilon_{Si} + \sum_i (a_i - \mathbb{E}_n a)(2\beta_0 - \widehat{\beta}_0 - \alpha_0) \right].$$

We use $\sum_i x_i$, $\sum_i y_i$ and $\sum_i z_i$ to denote for shorthand $2\sum_i (a_i - \mathbb{E}_n a)\epsilon_{Di}$, $\sum_i (a_i - \mathbb{E}_n a)\epsilon_{Si}$ and $\sum_i (a_i - \mathbb{E}_n a)(2\beta_0 - \widehat{\beta}_0 - \alpha_0)$ respectively.

From Lemmas E.3 to E.5, we know that with probability at least $1 - 3\delta$, it holds that

$$|\frac{1}{n}\sum_i x_i| \leq \sqrt{\frac{2\nu^2 \log(\frac{2\log_2 T}{\delta})}{n}} + \frac{2\sigma_S \sigma_D \log(\frac{2\log_2 T}{\delta})}{n},$$

where $\nu$ are defined in Lemma E.3.

For $\frac{1}{n}\sum_i y_i$, we have the following lemma to illustrate its property.

**Lemma E.6.** *There exists constant $n_2$ such that with probability at least $1 - 3\delta$, it holds that for any $n \geq n_2$ where $n = 2^m$ and $m \in [\log_2 T]$,*

$$|\frac{1}{n}\sum_{i=1}^n y_i + \frac{\sigma_S^2}{2}| \leq \frac{3\bar{P}}{2}\sqrt{\frac{2\sigma_S^2 \log(\frac{2\log_2 T}{\delta})}{n}} + \sqrt{\frac{8\log(\frac{2\log_2 T}{\delta})}{n}}.$$

*Proof.* Let's decompose $y_i$ as $-\mathbb{E}_n a * \epsilon_{Si} + \frac{\widehat{\beta}_0 - \alpha_0}{2}\epsilon_{Si} - \frac{\epsilon_{Si}^2}{2}$.

For $-\mathbb{E}_n a * \epsilon_{Si}$, similar to Lemma E.4, it holds that with probability at least $1 - \delta$, for all $n = 2^m$,

$$|\frac{1}{n}\sum_i -\mathbb{E}_n a * \epsilon_{Si}| = |\frac{1}{n}\sum_i \epsilon_{Si}| * |\mathbb{E}_n a|$$

$$\leq \bar{P}|\frac{1}{n}\sum_i \epsilon_{Si}|$$

$$\leq \bar{P}\sqrt{\frac{2\sigma_S^2 \log(\frac{2\log_2 T}{\delta})}{n}}.$$

The first equation holds due to simple algebra while the first inequality holds because the service fee has a trivial upper bound $\bar{P}$. The last inequality holds due to Lemma D.2.

As for term $\frac{\widehat{\beta}_0 - \alpha_0}{2}\epsilon_{Si}$, we use $t_i$ to denote it. Like the proof of Lemma E.3, we know that $\widehat{\beta}_0$ only depends on previous $\epsilon_D$ and $\epsilon_S$ but don't $\epsilon_{Si}$. So, it holds that $\mathbb{E}[t_i \mid t_1, ..., t_{i-1}] = 0$ and we use $\mathcal{F}$ to denote the corresponding filtration. Since we have the following MGF

$$\mathbb{E}[e^{\lambda t_i} \mid \mathcal{F}_{i-1}] = e^{\frac{\lambda^2 \sigma_S^2 (\widehat{\beta}_0 - \alpha_0)}{32}} \leq e^{\frac{\lambda^2 \sigma_S^2 \bar{P}^2}{32}},$$

we know that $t_i$ is a sub-Gaussian random variable with a parameter $\frac{\sigma_S \bar{P}}{4}$, which is also sub-exponential with parameters $(\frac{\sigma_S \bar{P}}{4}, 0)$. With Lemma D.1, we know that with probability $1 - \delta$

$$|\frac{1}{n}\sum_i \frac{\widehat{\beta}_0 - \alpha_0}{2}\epsilon_{Si}| \leq \bar{P}\sqrt{\frac{\sigma_S^2 \log(\frac{2\log_2 T}{\delta})}{2n}} \text{ for all } m \in [\log_2 T] \text{ and } n = 2^m.$$

For the term $\frac{\sigma_{Si}^2}{2}$, since $\frac{\epsilon_{Si}^2}{\sigma_S^2} \sim \chi_1^2$ and $\chi_1^2$ is sub-exponential with parameters $(2, 4)$, we know that with probability $1 - \delta$,

$$|\frac{1}{n}\sum_i \frac{\epsilon_{Si}^2}{2} - \frac{\sigma_S^2}{2}| \leq \sqrt{\frac{8\sigma_S^4 \log(\frac{2\log_2 T}{\delta})}{n}},$$

as long as $n \geq 8\log(\frac{2\log_2 T}{\delta}) := n_2$.

Accordingly, combining the above three terms leads to the needed lemma. $\qquad \square$

For $\frac{1}{n}\sum_i z_i$, we notice that it's indeed zero. It holds that

$$|\frac{1}{n}\sum_i z_i| = |\frac{1}{n}\sum_i (a_i - \mathbb{E}_n a)(2\beta_0 - \widehat{\beta}_0 - \alpha_0)| = 0,$$

due to the definition of $\mathbb{E}_n a$.

With the bounds for both nominator and denominator, it holds that

$$|\widehat{\beta}_1 - \beta_1|$$

$$\leq \frac{2(\alpha_1 - \beta_1)(\frac{1}{2}\sqrt{\frac{2\nu^2 \log(\frac{2\log_2 T}{\delta})}{n}} + \frac{\sigma_S \sigma_D \log(\frac{2\log_2 T}{\delta})}{n})}{\frac{\sigma_S^2}{2} - \sqrt{\frac{2\nu^2 \log(\frac{2\log_2 T}{\delta})}{n}} - \frac{2\sigma_S \sigma_D \log(\frac{2\log_2 T}{\delta})}{n} - \frac{3\bar{P}}{2}\sqrt{\frac{2\sigma_S^2 \log(\frac{2\log_2 T}{\delta})}{n}} - \sqrt{\frac{8\sigma_S^4 \log(\frac{2\log_2 T}{\delta})}{n}}}.$$

Then, there exists constants $C_2$, $C_3$ and $C_4$ such that when $n \geq \max\{n_1, n_2, C_3 \log\log T\} = C_4 \log\log T$, $|\widehat{\beta}_1 - \beta_1| \leq \frac{C_2 \sqrt{\log\log T}}{\sigma_S \sqrt{n}}$ due to $\nu = \sqrt{2}\sigma_S \sigma_D$. Note that $C_4 \lesssim \mathcal{O}(\frac{1}{\sigma_S^2})$ from previous definitions and process of proof.

When $n \leq C_4 \log\log T$, we can choose large enough constant $C_5$ such that $\frac{C_5}{\sigma_S \sqrt{C_4}}$ is greater than a trivial bound for $\beta_1$. Since we care about the situation when $\sigma_S$ is small, the probable appearance of it in the denominator won't cause trouble for us. Afterwards, by setting $C_1 = \max\{C_2, C_5\}$, we find a uniform constant such that

$$|\widehat{\beta}_1 - \beta_1| \leq \frac{C_1 \sqrt{\log\log T}}{\sigma_S \sqrt{2^m}}.$$

Moreover, we know that after the $m$-th episode, i.e. in the $m + 1$-th episode, $t \in \{2^{m+1} - 1 : 2^{m+2} - 1\}$, then $t \approx \Theta(2^m)$. Therefore, from the above, we know

$$|\widehat{\beta}_{1t} - \beta_1| \lesssim \mathcal{O}(\frac{\sqrt{\log\log T}}{\sigma_S \sqrt{t}}) \text{ for all } t \in [T].$$

Whereas when calculating the probability of bad events, there is some redundancy, after calibration, the union bound on the probability of bad events is $6\delta$ which ends the proof. $\qquad \square$

Since we can estimate $\beta_1$ with high accuracy, we similarly have the following proposition about the estimates of $\beta_0$ when Proposition E.2 holds.

**Proposition E.7.** *Let $\widehat{\beta}_0$ be the estimated intercept of Algorithm 4 after the $m$-th episode. Then, under Assumption 2.1 and conditional on the event that Proposition E.2 holds, there exists a constant $D_1$ such that with probability at least $1 - \delta$, we have*

$$|\widehat{\beta}_0 - \beta_0| \leq \frac{D_1 \sqrt{\log\log T}}{\sigma_S \sqrt{2^m}} \text{ for all } m \in [\log_2 T],$$

*then*

$$|\widehat{\beta}_{0t} - \beta_0| \lesssim \mathcal{O}(\frac{\sqrt{\log\log T}}{\sigma_S \sqrt{t}}) \text{ for all } t \in [T],$$

*where subscript $t$ represents which round.*

*Proof.* It holds that for any $n = 2^m$, with probability at least $1 - \delta$,

$$|\widehat{\beta}_0 - \beta_0| = |\beta_1 \mathbb{E}_n Q^e - \widehat{\beta}_1 \mathbb{E}_n Q^e + \mathbb{E}_n \epsilon_D|$$

$$\leq |\beta_1 \mathbb{E}_n Q^e - \widehat{\beta}_1 \mathbb{E}_n Q^e| + |\mathbb{E}_n \epsilon_D|$$

$$= |\mathbb{E}_n Q^e| * |\beta_1 - \widehat{\beta}_1| + |\mathbb{E}_n \epsilon_D|$$

$$\leq \frac{C_1 \bar{Q} \sqrt{\log\log T}}{\sigma_S \sqrt{n}} + \sqrt{\frac{2\sigma_D^2 \log(\frac{2\log_2 T}{\delta})}{n}}.$$

The first equation comes from Algorithm 4. The first inequality holds due to the triangle inequality while the second inequality holds due to Proposition E.2 and Lemma E.5.

Then, for any $n = 2^m$, there exists a $D_1$ such that $|\widehat{\beta}_0 - \beta_0| \leq \frac{D_1 \sqrt{\log\log T}}{\sigma_S \sqrt{n}}$. Similarly, since $t \approx \Theta(n)$, it holds that

$$|\widehat{\beta}_{0t} - \beta_0| \lesssim \mathcal{O}(\frac{\sqrt{\log\log T}}{\sigma_S \sqrt{t}}).$$

Finally, bad events coming from $\mathbb{E}_n \epsilon_D$ have a cumulative probability of $\delta$ which closes the proof. $\quad \square$

Up to now, we provide proof of Propositions E.2 and E.7 showing that even when IV is not i.i.d. with respect to time, we can obtain good estimation results, demonstrating the effectiveness of our Algorithm 4.

### E.1.2 Proof of Theorem E.1

*Proof.* With Appendix E.1.1 in hand, we are ready to give a high probability upper bound for the total regret.

We use $f(a)$ to denote $\mathbb{E}a * Q^e(P_S, P_D, a) = \frac{\beta_0 - \alpha_0 - a - \epsilon_S}{\alpha_1 - \beta_1} a$. Then, with an ordinary Taylor-series expansion and the mean value theorem, it holds that

$$\text{SubOpt}_t(a_t) = \frac{df(a_t^*)}{da}(a_t^* - a_t) + \frac{1}{2}\frac{d^2 f(\alpha a_t^* + (1 - \alpha)a_t)}{da^2}(a_t^* - a_t)^2,$$

where $\alpha \in [0, 1]$

From Lemma E.1, we know that $\frac{df(a_t^*)}{da} = 0$. Besides, it holds that $(a_t^* - a_t)^2 \lesssim \mathcal{O}(|\widehat{\beta}_{0t} - \beta_0|^2) \lesssim \mathcal{O}(\frac{\log\log T}{\sigma_S^2 t})$ due to Proposition E.7. Then, since $|\frac{d^2 f}{da^2}|$ is uniformly bounded by a constant, namely $\frac{2}{\alpha_1 - \beta_1}$, we have that

$$\text{SubOpt}_t(a_t) \lesssim \mathcal{O}(\frac{\log\log T}{\sigma_S^2 t}).$$

As a result, it holds that

$$\text{Regret}(T) = \sum_{t=1}^T \text{SubOpt}_t(a_t) \lesssim \sum_{t=1}^T \mathcal{O}(\frac{\log\log T}{\sigma_S^2 t}) \lesssim \mathcal{O}(\frac{\log T \log\log T}{\sigma_S^2}).$$

The last inequality holds due to $\sum_{i=1}^n \frac{1}{i} \leq 1 + \log n$.

The total odds of some inequalities not holding is $7\delta$. Among them, $6\delta$ derives from Proposition E.2 and $\delta$ originates from Proposition E.7. Note that $\delta$ only appears in $\log\log T$ terms. Setting $\iota = 7\delta$ ends our proof. $\square$

We then turn to prove Theorem 3.2. We begin this section with some preparing lemmas. Then, we will prove Theorem 3.2 in leverage of them. Finally, we will give a brief proof of the statement on time-varying discount.

### E.1.3 Useful Facts for Proving Theorem 3.2

Since the buyer may behave as having a different demand curve $P_D'$ rather than the true one $P_D$, we first give the following proposition of depicting such untruthfulness.

**Proposition E.8.** *For any behavior $P_D'$, we can fully characterize it with a drift parameter $\eta_D$. Consequently, without loss of generality, we can assume that*

$$P_{Dt}' = \beta_0 + \beta_1 Q + \epsilon_{Dt} + \eta_{Dt}.$$

*Proof.* Since the platform can only observe achieved equilibrium price $P_t^e$ and quantity $Q_t^e$ each round, we only need to find an associated $\eta_{Dt}$ leading to the same results.

By setting $\eta_{Dt} = P_t^e - \beta_0 - \beta_1 Q_t^e - \epsilon_{Dt}$, we find that $(P_t^e, Q_t^e)$ will satisfy Equation (1). Since two lines in Euclidean space have at most one intersection point, we know that $\eta_{Dt}$ fully describes $P_{Dt}'$, and this correspondence is unique. Hence, we can use $\eta_D$ to represent untruthful behavior $P_D'$ and it finishes our proof. $\square$

Using the equivalence in Proposition E.8, we present the relationship between equilibrium price and quantity and the service fee.

**Lemma E.9.** *For any valid service fee $a$ and tuple $(P_S, P_D')$, the equilibrium price and quantity follow*

$$Q^e = \frac{\beta_0 - \alpha_0 - a + \epsilon_D - \epsilon_S + \eta_D}{\alpha_1 - \beta_1}$$

*and*

$$P^e = \frac{\alpha_1\beta_0 - \alpha_0\beta_1 - \beta_1 a + \alpha_1\epsilon_D - \beta_1\epsilon_S + \alpha_1\eta_D}{\alpha_1 - \beta_1},$$

*respectively. Then, the change coming from $\eta_D$ leads to*

$$\Delta Q^e = \frac{\eta_D}{\alpha_1 - \beta_1} \text{ and } \Delta P^e = \frac{\alpha_1\eta_D}{\alpha_1 - \beta_1}.$$

*Proof.* From Proposition E.8 and Equation (1), we can have the formulas for $Q^e$ and $P^e$ with some simple algebra. Furthermore, comparing them and the ones without $\eta_D$ leads to the formulas about $\Delta Q^e$ and $\Delta P^e$, which finishes the proof. $\qquad\square$

With these preparations, we are ready to give some properties of $\eta_{Dt}$ and show good performance of Algorithm 1.

**Lemma E.10.** *Considering the $m$-th episode of Algorithm 1, in the $i$-th round of this episode, we use $\eta_{Di}$ to denote $\eta_{D(2^m+i-2)}$ though a little abuse of notations. Then, it holds that for any rational buyer,*

$$|\eta_{Di}| \leq \frac{(\alpha_1 - \beta_1)\bar{Q}}{\sqrt{1-\gamma}}\gamma^{\frac{n-i+1}{2}},$$

*where the $m$-th episode contains $n = 2^m$ rounds. More precisely, it holds that*

$$\sum_{i=1}^{n} \gamma^{i-n-1}\eta_{Di}^2 \leq \frac{(\alpha_1 - \beta_1)^2\bar{Q}^2}{1-\gamma}.$$

*Proof.* From Assumption 2.2, we know that

$$\text{Sur} = \int_0^{Q^e} (P_D - P_S - a)dQ.$$

Therefore, we know that in the $i$-th round of the $m$-th episode, the loss of surplus is

$$\Delta\text{Sur}_i = \frac{1}{2}(\Delta Q_i^e)^2(\beta_1 - \alpha_1) = -\frac{1}{2}\frac{\eta_{Di}^2}{\alpha_1 - \beta_1}.$$

Since the update of the estimation of $(\beta_0, \beta_1)$ happens in $n - i$ rounds, the maximum gain from misreport is bounded by

$$\sum_{j=n-i}^{T} \gamma^{j+1}\frac{1}{2}(\alpha_1 - \beta_1)\bar{Q}^2 \leq \sum_{j=n-i}^{\infty} \gamma^{j+1}\frac{1}{2}(\alpha_1 - \beta_1)\bar{Q}^2 = \frac{\gamma^{n-i+1}(\alpha_1 - \beta_1)\bar{Q}^2}{2(1-\gamma)},$$

because the surplus is bounded by a trivial bound $\frac{1}{2}(\alpha_1 - \beta_1)\bar{Q}^2$ in each round. The inequality holds because every term is positive. With some calculations, we have the final equation.

Hence, we know that if the buyer is rational, she will guarantee that

$$-\frac{1}{2}\frac{\eta_{Di}^2}{\alpha_1 - \beta_1} + \frac{\gamma^{n-i+1}(\alpha_1 - \beta_1)\bar{Q}^2}{2(1-\gamma)} \geq 0,$$

ending with

$$|\eta_{Di}| \leq \frac{(\alpha_1 - \beta_1)\bar{Q}}{\sqrt{1-\gamma}}\gamma^{\frac{n-i+1}{2}}.$$

When we consider all $n$ rounds in the $m$-th episode as a whole, we know the total loss of surplus at the point of the last round in the $m-1$-th episode is $\sum_{i=1}^{n} \gamma^i \frac{1}{2}\frac{\eta_{Di}^2}{\alpha_1-\beta_1}$. The maximum surplus increment is less than $\gamma^{n+1}\frac{(\alpha-\beta_1)\bar{Q}^2}{2(1-\gamma)}$. For a rational and strategic buyer, it holds that

$$\sum_{i=1}^{n} \gamma^i \frac{1}{2}\frac{\eta_{Di}^2}{\alpha_1 - \beta_1} \leq \gamma^{n+1}\frac{(\alpha - \beta_1)\bar{Q}^2}{2(1-\gamma)},$$

leading to

$$\sum_{i=1}^{n} \gamma^{i-n-1} \eta_{Di}^2 \le \frac{(\alpha_1 - \beta_1)^2 \bar{Q}^2}{1 - \gamma},$$

which finishes our proof.

$\square$

We notice that since the buyer owns strictly less than 1 discount rate, if we inverse time, the bound of misreport will exponentially decay. Therefore, we have reason to believe that these misreports will not significantly affect the estimation of parameters. Consequently, we have the following propositions.

**Proposition E.11.** *Let $\widehat{\beta}_1$ be the estimated slope of Algorithm 4 after the $m$-th episode. Then, under Assumptions 2.1 and 2.2, with probability at least $1 - 6\delta$, it holds that when $2^m \gtrsim \mathcal{O}(\frac{\log \log T}{\sigma_S^2} + \frac{1}{\sigma_S^2(1-\gamma)})$,*

$$|\widehat{\beta}_1 - \beta_1| \lesssim \mathcal{O}(\frac{\sqrt{\log \log T}}{\sigma_S \sqrt{2^m}} + \frac{1}{2^m \sigma_S^2(1-\gamma)}) \text{ for all } m \in [\log_2 T],$$

*then*

$$|\widehat{\beta}_{1t} - \beta_1| \lesssim \mathcal{O}(\frac{\sqrt{\log \log T}}{\sigma_S \sqrt{t}} + \frac{1}{\sigma_S^2(1-\gamma)t}) \text{ for all } t \in [T],$$

*where subscript $t$ represents which round.*

*Proof.* Similar to Proposition E.2, it holds that

$$\widehat{\beta}_1 = \beta_1 + \frac{\frac{1}{n}\sum_{i=1}^{n} \epsilon_{Di}(a_i - \mathbb{E}_n a)}{\frac{1}{n}\sum_{i=1}^{n} Q_i^e(a_i - \mathbb{E}_n a)} + \frac{\frac{1}{n}\sum_{i=1}^{n} \eta_{Di}(a_i - \mathbb{E}_n a)}{\frac{1}{n}\sum_{i=1}^{n} Q_i^e(a_i - \mathbb{E}_n a)}.$$

For the numerator, we know that with probability at least $1 - 3\delta$,

$$|\frac{1}{n}\sum_{i=1}^{n} \epsilon_{Di}(a_i - \mathbb{E}_n a)| \le \frac{1}{2}\sqrt{\frac{2\nu^2 \log(\frac{2\log_2 T}{\delta})}{n}} + \frac{\sigma_S \sigma_D \log(\frac{2\log_2 T}{\delta})}{n},$$

due to Lemmas E.3 to E.5.

Moreover, we bound $\frac{1}{n}\sum_{i=1}^{n} \eta_{Di}(a_i - \mathbb{E}_n a)$ directly in leverage of Lemma E.10. It holds that

$$\begin{aligned}
|\frac{1}{n}\sum_{i=1}^{n} \eta_{Di}(a_i - \mathbb{E}_n a)| &\le \frac{2\bar{P}}{n}\sum_{i=1}^{n} |\eta_{Di}| \\
&\le \frac{2\bar{P}}{n}\sqrt{\sum_{i=1}^{n} \gamma^{i-n-1}\eta_{Di}^2}\sqrt{\sum_{i=1}^{n} \gamma^{n+1-i}} \\
&\le \frac{2(\alpha_1 - \beta_1)\bar{P}\bar{Q}}{n\sqrt{1-\gamma}}\sqrt{\frac{\gamma}{1-\gamma}} \\
&= \frac{2(\alpha_1 - \beta_1)\bar{P}\bar{Q}\sqrt{\gamma}}{n(1-\gamma)}.
\end{aligned}$$

The first inequality holds due to $a_i \le \bar{P}$ while the second inequality holds due to Lemma E.10 and Cauchy–Schwarz inequality. The third inequality holds because every term in the summation is positive and the geometric series sum formula.

For the denominator, since $\Delta Q^e = \eta_D/(\alpha_1 - \beta_1)$ from Lemma E.9, we only need to bound $\frac{1}{n}\sum_{i=1}^{n} \frac{\eta_D}{\alpha_1 - \beta_1}(a_i - \mathbb{E}_n a)$. It holds that

$$\frac{1}{n}\sum_{i=1}^{n} \frac{\eta_D}{\alpha_1 - \beta_1}(a_i - \mathbb{E}_n a) \le \frac{2\bar{P}\bar{Q}\sqrt{\gamma}}{n(1-\gamma)}$$

due to the same process as bounding $|\frac{1}{n}\sum_{i=1}^{n} \eta_{Di}(a_i - \mathbb{E}_n a)|$ above.

Therefore, we notice that

$$|\frac{1}{n}\sum_{i=1}^n Q_i^e(a_i - \mathbb{E}_n a)| \geq \frac{1}{2(\alpha_1 - \beta_1)}\left(\frac{\sigma_S^2}{2} - \sqrt{\frac{2\nu^2 \log(\frac{2\log_2 T}{\delta})}{n}} - \frac{2\sigma_S \sigma_D \log(\frac{2\log_2 T}{\delta})}{n}\right.$$

$$\left. - \frac{3\bar{P}}{2}\sqrt{\frac{2\sigma_S^2 \log(\frac{2\log_2 T}{\delta})}{n}} - \sqrt{\frac{8\sigma_S^4 \log(\frac{2\log_2 T}{\delta})}{n}}\right) - \frac{2\bar{P}\bar{Q}\sqrt{\gamma}}{n(1-\gamma)},$$

with probability at least $1 - 6\delta$ due to Lemma E.6.

Combining all these terms, it holds that

$$|\widehat{\beta}_1 - \beta_1| \leq \left|\frac{\frac{1}{n}\sum_{i=1}^n \epsilon_{Di}(a_i - \mathbb{E}_n a)}{\frac{1}{n}\sum_{i=1}^n Q_i^e(a_i - \mathbb{E}_n a)}\right| + \left|\frac{\frac{1}{n}\sum_{i=1}^n \eta_{Di}(a_i - \mathbb{E}_n a)}{\frac{1}{n}\sum_{i=1}^n Q_i^e(a_i - \mathbb{E}_n a)}\right|$$

$$\lesssim \frac{\mathcal{O}(\sigma_S \frac{\sqrt{\log\log T}}{\sqrt{n}})}{\mathcal{O}(\sigma_S^2) - \mathcal{O}(\sigma_S \frac{\sqrt{\log\log T}}{\sqrt{n}}) - \mathcal{O}(\frac{1}{n(1-\gamma)})} + \frac{\mathcal{O}(\frac{1}{n(1-\gamma)})}{\mathcal{O}(\sigma_S^2) - \mathcal{O}(\sigma_S \frac{\sqrt{\log\log T}}{\sqrt{n}}) - \mathcal{O}(\frac{1}{n(1-\gamma)})}$$

$$\lesssim \mathcal{O}(\frac{\sqrt{\log\log T}}{\sigma_S \sqrt{n}} + \frac{1}{n\sigma_S^2(1-\gamma)})$$

$$\lesssim \mathcal{O}(\frac{\sqrt{\log\log T}}{\sigma_S \sqrt{2^m}} + \frac{1}{2^m \sigma_S^2(1-\gamma)}).$$

The first inequality holds due to the triangle inequality while the second inequality holds due to previously presented concentration bounds. The third inequality holds due to simple algebra while the last one holds due to $n = 2^m$. Besides, taking into account repeated bad events, the probability of all inequalities holding is at least $1 - 6\delta$. Here, we need $n \geq \mathcal{O}(\frac{\log\log T}{\sigma_S^2} + \frac{1}{\sigma_S^2(1-\gamma)})$.

Considering that for any round in the $m + 1$-th episode, we have $t \in \{2^{m+1} - 1 : 2^{m+2} - 1\}$, it then holds that

$$|\widehat{\beta}_{1t} - \beta_1| \lesssim \mathcal{O}(\frac{\sqrt{\log\log T}}{\sigma_S \sqrt{t}} + \frac{1}{\sigma_S^2 t(1-\gamma)}) \text{ for all } t \in [T],$$

which ends the proof. $\qquad\square$

Conditional on having a good estimation of $\beta_1$, we have the following proposition, saying that we can also estimate $\beta_0$ precisely.

**Proposition E.12.** *Let $\widehat{\beta}_0$ be the estimated intercept of Algorithm 4 after the $m$-th episode. Then, under Assumptions 2.1 and 2.2 and conditional on the event that Proposition E.11 holds, we have with probability at least $1 - \delta$,*

$$|\widehat{\beta}_0 - \beta_0| \lesssim \mathcal{O}(\frac{\sqrt{\log\log T}}{\sigma_S \sqrt{2^m}} + \frac{1}{2^m \sigma_S^2(1-\gamma)}) \text{ for all } m \in [\log_2 T],$$

*then*

$$|\widehat{\beta}_{0t} - \beta_0| \lesssim \mathcal{O}(\frac{\sqrt{\log\log T}}{\sigma_S \sqrt{t}} + \frac{1}{\sigma_S^2(1-\gamma)t}) \text{ for all } t \in [T],$$

*where subscript $t$ represents which round.*

*Proof.* We know that $\widehat{\beta}_0 = \mathbb{E}_n P^e - \widehat{\beta}_1 \mathbb{E}_n Q^e$. Thus, it holds that

$$
\begin{aligned}
|\widehat{\beta}_0 - \beta_0| &\leq |\mathbb{E}_n P^e - \widehat{\beta}_1 \mathbb{E}_n Q^e - \beta_0| \\
&\leq |\beta_1 \mathbb{E}_n Q^e + \mathbb{E}_n \epsilon_D + \mathbb{E}_n \eta_D - \widehat{\beta}_1 \mathbb{E}_n Q^e| \\
&\leq |\widehat{\beta}_1 - \beta_1| * |\mathbb{E}_n Q^e| + |\mathbb{E}_n \epsilon_D| + |\mathbb{E}_n \eta_D| \\
&\leq |\widehat{\beta}_1 - \beta_1| * \bar{Q} + |\mathbb{E}_n \epsilon_D| + \mathbb{E}_n |\eta_D| \\
&\lesssim \bar{Q} * \mathcal{O}\left(\frac{\sqrt{\log\log T}}{\sigma_S \sqrt{n}} + \frac{1}{n\sigma_S^2(1-\gamma)}\right) + \sqrt{\frac{2\sigma_D^2 \log(\frac{2\log_2 T}{\delta})}{n}} + \frac{(\alpha_1 - \beta_1)\bar{Q}\sqrt{\gamma}}{n(1-\gamma)} \\
&\lesssim \mathcal{O}\left(\frac{\sqrt{\log\log T}}{\sigma_S \sqrt{n}} + \frac{1}{n\sigma_S^2(1-\gamma)}\right).
\end{aligned}
$$

The first inequality holds due to the definition of $\widehat{\beta}_0$ while the second inequality holds due to Proposition E.8. The third inequality holds due to the triangle inequality while the fourth inequality holds due to Jensen's inequality and trivial upper bound $\bar{Q}$. The fifth inequality holds with probability at least $1 - \delta$ because of the results achieved in the proof of Proposition E.11 while the last inequality holds due to simple algebra.

Similarly, we have that

$$
|\widehat{\beta}_{0t} - \beta_0| \lesssim \mathcal{O}\left(\frac{\sqrt{\log\log T}}{\sigma_S \sqrt{t}} + \frac{1}{\sigma_S^2 t(1-\gamma)}\right) \text{ for all } t \in [T],
$$

which ends the proof. $\qquad\square$

Propositions E.11 and E.12 highlight the statistical efficiency of Algorithm 1 even when the buyer is far-sighted, showing the robustness of our algorithm against non-myopic strategic agents.

### E.1.4 Proof of Theorem 3.2

*Proof.* We recall the definition of $\text{SubOpt}_t$ in round $t$ first, that is,

$$
\text{SubOpt}_t(a_t) = \mathbb{E}[a_t^* * Q^e(P_{St}, P_{Dt}, a_t^*) - a_t * Q^e(P_{St}, P'_{Dt}, a_t)].
$$

Hence, we can use the triangle inequality to gain

$$
\begin{aligned}
\text{SubOpt}_t&(a_t) \\
&\leq \mathbb{E}|a_t^* * Q^e(P_{St}, P_{Dt}, a_t^*) - a_t * Q^e(P_{St}, P_{Dt}, a_t)| + \\
&\quad \mathbb{E}|a_t * Q^e(P_{St}, P_{Dt}, a_t) - a_t * Q^e(P_{St}, P'_{Dt}, a_t)|.
\end{aligned}
$$

For the first term, it holds that

$$
\mathbb{E}|a_t^* * Q^e(P_{St}, P_{Dt}, a_t^*) - a_t * Q^e(P_{St}, P_{Dt}, a_t)| \lesssim \mathcal{O}\left(\frac{\log\log T}{\sigma_S^2 t} + \frac{1}{\sigma_S^2 t(1-\gamma)}\right)
$$

where the first term is similar to the proof of Theorem E.1. For the second term, since $|\frac{df}{da}| \leq \frac{2\bar{P}}{(\alpha_1 - \beta_1)}$, we use the first-order approximation to obtain it. We thereafter know that

$$
\sum_{t=1}^{T} \mathbb{E}|a_t^* * Q^e(P_{St}, P_{Dt}, a_t^*) - a_t * Q^e(P_{St}, P_{Dt}, a_t)| \lesssim \mathcal{O}\left(\frac{\log T \log\log T}{\sigma_S^2} + \frac{\log T}{\sigma_S^2(1-\gamma)}\right).
$$

For the second term, it holds that

$$
\begin{aligned}
\mathbb{E}|a_t Q^e(P_{St}, P_{Dt}, a_t) - a_t * Q^e(P_{St}, P'_{Dt}, a_t)| &\leq \mathbb{E}|a_t| * |Q^e(P_{St}, P_{Dt}, a_t) - Q^e(P_{St}, P'_{Dt}, a_t)| \\
&\leq \bar{P} * \mathbb{E}|Q^e(P_{St}, P_{Dt}, a_t) - Q^e(P_{St}, P'_{Dt}, a_t)| \\
&\leq \frac{\bar{P}}{\alpha_1 - \beta_1}|\eta_{Dt}|.
\end{aligned}
$$

The first inequality holds due to simple algebra while the second inequality holds because the service fee is always no larger than $\bar{P}$. The last inequality holds due to Lemma E.9.

Then, during the $m$-th episode, there are $n = 2^m$ rounds and we have

$$\frac{\bar{P}}{\alpha_1 - \beta_1}|\eta_{Di}| \leq \frac{\bar{P}\bar{Q}}{\sqrt{1-\gamma}}(\sqrt{\gamma})^{n-i+1},$$

and

$$\sum_{i=1}^{n} \gamma^{i-n-1}\eta_{Di}^2 \leq \frac{(\alpha_1 - \beta_1)^2\bar{Q}^2}{1-\gamma}.$$

Furthermore, it holds that

$$\sum_{t=1}^{T}\frac{\bar{P}}{\alpha_1 - \beta_1}|\eta_{Dt}| = \sum_{m=1}^{\log_2 T}\sum_{i=1}^{2^m}\frac{\bar{P}}{\alpha_1 - \beta_1}|\eta_{Di}| \leq \sum_{m=1}^{\log_2 T}\frac{\bar{P}}{\alpha_1 - \beta_1}\frac{(\alpha_1 - \beta_1)\bar{Q}\sqrt{\gamma}}{1-\gamma} = \frac{\bar{P}\bar{Q}\sqrt{\gamma}\log_2 T}{1-\gamma}.$$

The inequality holds due to the same process when proving Proposition E.11. It comes from Lemma E.10 and Cauchy-Schwartz inequality.

Consequently, we have

$$\text{Regret}(T) = \sum_{t=1}^{T}\text{SubOpt}_t(a_t)$$

$$\lesssim \mathcal{O}(\frac{\log T\log\log T}{\sigma_S^2} + \frac{\log T}{\sigma_S^2(1-\gamma)}) + \frac{\bar{P}\bar{Q}\sqrt{\gamma}\log_2 T}{1-\gamma} + \mathcal{O}(\frac{\log\log T}{\sigma_S^2} + \frac{1}{\sigma_S^2(1-\gamma)})$$

$$\lesssim \mathcal{O}(\frac{\log T\log\log T}{\sigma_S^2} + \frac{\log T}{\sigma_S^2(1-\gamma)}),$$

with probability at least $1 - 7\delta$. For the first inequality, the first term comes from suboptimal service fees while the second term comes from misreport. The last term is needed because of the requirement of $n \geq \mathcal{O}(\frac{\log\log T}{\sigma_S^2} + \frac{1}{\sigma_S^2(1-\gamma)})$.

Finally, as $\delta$ merely hides within $\log\log T$ terms, setting $\iota = 7\delta$ leads to the desired result and finishes our proof. □

### E.1.5 When Facing Time-varying Discount

When the buyer has a time-varying discount rate $\gamma_t$, we can replace it with its uniform upper confidence bound $\bar{\gamma}$ in the proofs of Lemma E.10, propositions E.11 and E.12, and theorem 3.2. Since we use upper bound $\bar{\gamma}$ for every spot discount rate, all estimators become more conservative. Intuitively, the higher the discount rate, the greater the buyer's motivation to misreport. So, all inequalities will hold with corresponding probabilities.

Therefore, when the discount rates are changing in different rounds but with a uniform upper bound $\bar{\gamma}$, Algorithm 1 still achieves $\mathcal{O}(\log T\log\log T)$ regret with high probability which ends the proof.

### E.2 Omitted Proof in Section 3.1

So as to prove Theorem 3.1, we state some auxiliary lemmas at first.

### E.2.1 Useful Facts for Proving Theorem 3.1

Recall that we use $\epsilon$ to represent added noise which follows $\mathcal{N}(0, \frac{1}{\sqrt{\text{Size}(\mathcal{A})+1}})$ and $\eta_D$ to represent the distance between $P'_D$ and $P_D$ thanks to Lemma E.9. We have the following lemma to depict the equilibrium price and quantity in each round.

**Lemma E.13.** *For any estimation $\widehat{\beta}_0$ and tuple $(P_S, P'_D)$, the equilibrium price follows*

$$Q^e = \frac{2\beta_0 - \widehat{\beta}_0 - \alpha_0 + 2\epsilon_D + 2\eta_D - 2\epsilon}{2(\alpha_1 - \beta_1)}.$$

*Then, the change coming from $\eta_D$ leads to*

$$\Delta Q^e = \frac{\eta_D}{\alpha_1 - \beta_1}.$$

*Proof.* We get the formula of $Q^e$ from Equation (1) with $\epsilon_S = 0$ and the implementation of Algorithm 2. Moreover, due to Lemma E.1, we know the optimal ex-ante service fee $a^* = \frac{\beta_0 - \alpha_0}{2}$ without untruthful behavior and extra noise term $\epsilon$. $\qquad\square$

Since the choice of $\epsilon$ is independent of $\eta_D$, we know that Lemma E.10 still holds, that is, in the $i$-th round of $m$-th episode, the quantity of misreport $\eta_{Di}$ follows that for the rational and strategic buyer,

$$\sum_{i=1}^{n} \gamma^{i-n-1}\eta_{Di}^2 \le \frac{(\alpha_1 - \beta_1)^2 \bar{Q}^2}{1 - \gamma}, \tag{2}$$

where the $m$-th episode contains $n = 2^m$ rounds.

Subsequently, we are ready to bound the distance between the estimates of $\beta_0$ and $\beta_1$ and their true values.

**Proposition E.14.** *Let $\widehat{\beta}_1$ be the estimated slope of Algorithm 4 after the $m$-th episode. Then, under Assumptions 2.1 and 2.2, with probability at least $1 - 8\delta$, it holds that when $2^m \gtrsim \mathcal{O}(\log T \log \log T + \frac{1}{(1-\gamma)^2})$,*

$$|\widehat{\beta}_1 - \beta_1| \lesssim \mathcal{O}(\frac{\sqrt{\log \log T}}{2^{\frac{m}{4}}} + \frac{1}{2^{\frac{m}{2}}(1 - \gamma)}) \text{ for all } m \in [\log_2 T],$$

*then*

$$|\widehat{\beta}_{1t} - \beta_1| \lesssim \mathcal{O}(\frac{\sqrt{\log \log T}}{t^{\frac{1}{4}}} + \frac{1}{(1 - \gamma)\sqrt{t}}) \text{ for all } t \in [T],$$

*where subscript $t$ represents which round.*

*Proof.* Let's first decompose the error terms of estimating $\beta_1$. Assume that $n = 2^m$ as usual. From the implementation of Algorithm 4, we know that

$$\widehat{\beta}_1 = \frac{\mathbb{E}_n[(a - \mathbb{E}_n a)P^e]}{\mathbb{E}_n[(a - \mathbb{E}_n a)Q^e]} = \beta_1 + \left| \frac{\frac{1}{n}\sum_{i=1}^{n} \epsilon_{Di}(a_i - \mathbb{E}_n a)}{\frac{1}{n}\sum_{i=1}^{n} Q_i^e(a_i - \mathbb{E}_n a)} \right| + \left| \frac{\frac{1}{n}\sum_{i=1}^{n} \eta_{Di}(a_i - \mathbb{E}_n a)}{\frac{1}{n}\sum_{i=1}^{n} Q_i^e(a_i - \mathbb{E}_n a)} \right|.$$

Since $a = \frac{\widehat{\beta}_0 - \alpha_0}{2} + \epsilon$, it holds that $\frac{1}{n}\sum_{i=1}^{n} \epsilon_{Di}(a_i - \mathbb{E}_n a) = \frac{1}{n}\sum_{i=1}^{n} \epsilon_i * \epsilon_{Di} - \mathbb{E}_n \epsilon * \mathbb{E}_n \epsilon_D$. Let's bound them one by one.

First, let's give a high probability upper bound for $\frac{1}{n}\sum_{i=1}^{n} \epsilon_i * \epsilon_{Di}$. We note that it's a martingale difference sequence because $\epsilon_{Di}$ has mean zero conditional on previous events. From the proof of Lemma E.3, we know that $\epsilon_i * \epsilon_{Di}$ is sub-exponential with parameters $\nu_i = \sqrt{2}\frac{\sigma_D}{i^{1/4}}$ and $\alpha_i = 2\frac{\sigma_D}{i^{1/4}}$ because $\text{Var}(\epsilon_i) = \frac{1}{\sqrt{i}}$. Then, it holds that due to Lemma D.1,

$$\mathbb{P}(|\sum_{i=1}^{n} \epsilon_i * \epsilon_{Di}| \ge nt) \le 2\exp(-\frac{n^2 t^2}{4\sigma_D^2 \sum_{i=1}^{n} \sigma_i^2})$$

$$\le 2\exp(-\frac{n^2 t^2}{8\sigma_D^2 \sqrt{n}})$$

$$= 2\exp(-\frac{n^{3/2} t^2}{8\sigma_D^2}).$$

The first inequality holds due to Lemma D.1 while the second inequality holds due to $\sqrt{n} \le \sum_{i=1}^{n} \frac{1}{\sqrt{i}} \le 2\sqrt{n}$. Hence, by setting $t = \frac{\sqrt{8\sigma_D^2 \log(\frac{2\log_2 T}{\delta})}}{n^{3/4}}$, it holds that $\mathbb{P}(|\frac{1}{n}\sum_{i=1}^{n} \epsilon_i * \epsilon_{Di}| \ge t) \le \frac{\delta}{\log_2 T}$. Besides, we need $nt \le \frac{\sum_{i=1}^{n} \nu_i^2}{\max_i \alpha_i}$. A sufficient condition is $n \ge (8\log(\frac{2\log_2 T}{\delta}))^2$, then $n \gtrsim \mathcal{O}((\log \log T)^2)$. Finally, we have that with probability at least $1 - \delta$, for any $m \in [\log_2 T]$,

$$|\frac{1}{n}\sum_{i=1}^{n} \epsilon_{Di}\epsilon_i| \le \frac{\sqrt{8\sigma_D^2 \log(\frac{2\log_2 T}{\delta})}}{n^{3/4}}.$$

For the term $\mathbb{E}_n \epsilon_{Di}$, we have the following result from Lemma E.5

$$|\mathbb{E}_n \epsilon_{Di}| \leq \sqrt{\frac{2\sigma_D^2 \log(\frac{2 \log_2 T}{\delta})}{n}}$$

with probability at least $1 - \delta$.

For the term $\mathbb{E}_n \epsilon_i$, we know that $\epsilon_i$ is sub-Gaussian with parameter $\sigma_i = \frac{1}{i^{1/4}}$. Consequently, it holds that

$$|\mathbb{E}_n \epsilon_i| \leq \frac{\sqrt{4 \log(\frac{2 \log_2 T}{\delta})}}{n^{3/4}},$$

with cumulative probability at least $1 - \delta$ due to Lemma D.2.

We thereafter bound the term $\frac{1}{n} \sum_{i=1}^n \eta_{Di}(a_i - \mathbb{E}_n a)$. From Equation (2), we can obtain the following bound directly inherited from the proof of Proposition E.11 that

$$|\frac{1}{n} \sum_{i=1}^n \eta_{Di}(a_i - \mathbb{E}_n a)| \leq \frac{2(\alpha_1 - \beta_1)\bar{P}\bar{Q}\sqrt{\gamma}}{n(1 - \gamma)}.$$

After that, we need to investigate the property of $\sum_{i=1}^n Q_i^e(a_i - \mathbb{E}_n a)$. We divide it into three terms. It holds that

$$\frac{1}{n} \sum_{i=1}^n Q_i^e(a_i - \mathbb{E}_n a)$$

$$= \frac{1}{\alpha_1 - \beta_1}[\underbrace{\frac{1}{n} \sum_{i=1}^n \epsilon_{Di}(a_i - \mathbb{E}_n a)}_{q_1} + \underbrace{\frac{1}{n} \sum_{i=1}^n \eta_{Di}(a_i - \mathbb{E}_n a)}_{q_2} + \underbrace{\frac{1}{n} \sum_{i=1}^n (-\epsilon_i)(a_i - \mathbb{E}_n a)}_{q_3}].$$

From previous results, we know that with probability at least $1 - 3\delta$

$$|q_1| \leq \frac{\sqrt{8\sigma_D^2 \log(\frac{2 \log_2 T}{\delta})}}{n^{3/4}} + \frac{\sqrt{8\sigma_D^2 \log^2(\frac{2 \log_2 T}{\delta})}}{n^{5/4}},$$

and

$$|q_2| \leq \frac{2(\alpha_1 - \beta_1)\bar{P}\bar{Q}\sqrt{\gamma}}{n(1 - \gamma)}.$$

For the last term $q_3$, it holds that

$$-q_3 = \frac{1}{n} \sum_{i=1}^n \epsilon_i^2 - (\frac{1}{n} \sum_{i=1}^n \epsilon_i)^2.$$

Besides, we know that with probability at least $1 - \delta$, it holds that

$$(\frac{1}{n} \sum_{i=1}^n \epsilon_i)^2 \leq \frac{4 \log(\frac{2 \log_2 T}{\delta})}{n^{3/2}},$$

due to Lemma D.2.

Finally, for $\frac{1}{n} \sum_{i=1}^n \epsilon_i^2$, we know that $\epsilon_i \sim \mathcal{N}(0, \frac{1}{\sqrt{i}})$ and they are independent. Therefore, $\epsilon_i^2$ is sub-exponential random variable with $\nu_i = \frac{2}{\sqrt{i}}$ and $\alpha_i = \frac{4}{\sqrt{i}}$ with scaling of $\chi_1^2$ random variable.

So, from Lemma D.1, it holds that

$$\mathbb{P}(|\sum_{i=1}^n \epsilon_i^2 - \sum_{i=1}^n \frac{1}{\sqrt{i}}| \geq nt) \leq \max\left\{2 \exp(-\frac{n^2 t^2}{8 \sum_{i=1}^n \frac{1}{i}}), 2 \exp(-\frac{nt}{2\alpha^*})\right\},$$

where $\alpha^* = \max_i \alpha_i = 4$. By choosing $t = \max\left\{\frac{\sqrt{8(1+\log T)\log(\frac{2\log_2 T}{\delta})}}{n}, \frac{8\log(\frac{2\log_2 T}{\delta})}{n}\right\}$, we know that with probability at least $1 - \delta$, we have

$$\frac{1}{n}|\sum_{i=1}^{n}\epsilon_i^2 - \sum_{i=1}^{n}\frac{1}{\sqrt{i}}|$$

$$\leq \max\left\{\frac{\sqrt{8(1+\log T)\log(\frac{2\log_2 T}{\delta})}}{n}, \frac{8\log(\frac{2\log_2 T}{\delta})}{n}\right\} \lesssim \mathcal{O}(\frac{\sqrt{\log T \log\log T}}{n}),$$

for any $n = 2^m$. Here, we use facts $\log n \leq \sum_{i=1}^{n}\frac{1}{i} \leq 1 + \log n$ and $n \leq T$.

So, with probability at least $1 - 2\delta$, we know that

$$|q_3| \geq \frac{1}{\sqrt{n}} - \max\left\{\frac{\sqrt{8(1+\log T)\log(\frac{2\log_2 T}{\delta})}}{n}, \frac{8\log(\frac{2\log_2 T}{\delta})}{n}\right\} - \frac{4\log(\frac{2\log_2 T}{\delta})}{n^{3/2}}$$

$$\gtrsim \frac{1}{\sqrt{n}} - \mathcal{O}(\frac{\sqrt{\log T \log\log T}}{n}).$$

Combing all these terms, we know that

$$|\beta_1 - \widehat{\beta}_1| \leq \frac{(\alpha_1 - \beta_1)\left(\frac{\sqrt{8\sigma_D^2\log(\frac{2\log_2 T}{\delta})}}{n^{3/4}} + \frac{\sqrt{8\sigma_D^2\log^2(\frac{2\log_2 T}{\delta})}}{n^{5/4}} + \frac{2(\alpha_1-\beta_1)\bar{P}\bar{Q}\sqrt{\gamma}}{n(1-\gamma)}\right)}{\frac{1}{\sqrt{n}} - \mathcal{O}(\frac{\sqrt{\log T \log\log T}}{n}) - \frac{\sqrt{8\sigma_D^2\log(\frac{2\log_2 T}{\delta})}}{n^{3/4}} - \frac{\sqrt{8\sigma_D^2\log^2(\frac{2\log_2 T}{\delta})}}{n^{5/4}} - \frac{2(\alpha_1-\beta_1)\bar{P}\bar{Q}\sqrt{\gamma}}{n(1-\gamma)}}$$

$$\lesssim \mathcal{O}(\frac{\sqrt{\log\log T}}{n^{1/4}} + \frac{1}{\sqrt{n}(1-\gamma)}).$$

as long as $n \gtrsim \mathcal{O}(\log T \log\log T + \frac{1}{(1-\gamma)^2})$.

Since we know that $t \approx \Theta(n)$, it holds that with probability at least $1 - 8\delta$,

$$|\widehat{\beta}_{1t} - \beta_1| \lesssim \mathcal{O}(\frac{\sqrt{\log\log T}}{t^{\frac{1}{4}}} + \frac{1}{\sqrt{t}(1-\gamma)}),$$

which ends the proof. □

Now, we have the following proposition to bound the estimation error of $\beta_0$ in a similar manner.

**Proposition E.15.** *Let $\widehat{\beta}_0$ be the estimated intercept of Algorithm 4 after the $m$-th episode. Then, under Assumptions 2.1 and 2.2 and conditional on the event that Proposition E.14 holds, we have with probability at least $1 - \delta$,*

$$|\widehat{\beta}_0 - \beta_0| \lesssim \mathcal{O}(\frac{\sqrt{\log\log T}}{2^{\frac{m}{4}}} + \frac{1}{2^{\frac{m}{2}}(1-\gamma)}) \text{ for all } m \in [\log_2 T],$$

*then*

$$|\widehat{\beta}_{0t} - \beta_0| \lesssim \mathcal{O}(\frac{\sqrt{\log\log T}}{t^{\frac{1}{4}}} + \frac{1}{(1-\gamma)\sqrt{t}}) \text{ for all } t \in [T],$$

*where subscript $t$ represents which round.*

*Proof.* Similar to the proof of Proposition E.12, it holds that $\widehat{\beta}_0 = \mathbb{E}_n P^e - \widehat{\beta}_1 \mathbb{E}_n Q^e$. Consequently, we have with probability at least $1 - \delta$,

$$
\begin{aligned}
|\widehat{\beta}_0 - \beta_0| &\leq |\mathbb{E}_n P^e - \widehat{\beta}_1 \mathbb{E}_n Q^e - \beta_0| \\
&\leq |\beta_1 \mathbb{E}_n Q^e + \mathbb{E}_n \epsilon_D + \mathbb{E}_n \eta_D - \widehat{\beta}_1 \mathbb{E}_n Q^e| \\
&\leq |\widehat{\beta}_1 - \beta_1| * |\mathbb{E}_n Q^e| + |\mathbb{E}_n \epsilon_D| + |\mathbb{E}_n \eta_D| \\
&\leq |\widehat{\beta}_1 - \beta_1| * \bar{Q} + |\mathbb{E}_n \epsilon_D| + \mathbb{E}_n |\eta_D| \\
&\lesssim \bar{Q} * \mathcal{O}(\frac{\sqrt{\log \log T}}{n^{1/4}} + \frac{1}{\sqrt{n}(1-\gamma)}) + \sqrt{\frac{2\sigma_D^2 \log(\frac{2\log_2 T}{\delta})}{n}} + \frac{(\alpha_1 - \beta_1)\bar{Q}\sqrt{\gamma}}{n(1-\gamma)} \\
&\lesssim \mathcal{O}(\frac{\sqrt{\log \log T}}{n^{1/4}} + \frac{1}{\sqrt{n}(1-\gamma)}).
\end{aligned}
$$

The first four inequalities are the same as the one in the proof of Proposition E.12. Moreover, the fifth inequality comes from Proposition E.14 and the last inequality comes from simple order of magnitude analysis.

Similarly, we have that

$$
|\widehat{\beta}_{0t} - \beta_0| \lesssim \mathcal{O}(\frac{\sqrt{\log \log T}}{t^{\frac{1}{4}}} + \frac{1}{\sqrt{t}(1-\gamma)}) \text{ for all } t \in [T],
$$

which ends the proof. $\qquad\square$

Propositions E.14 and E.15 show that although we add approaching zero noise to explore the unknown environment, we can asymptotically estimate true parameters $\beta_0$ and $\beta_1$ accurately. Nevertheless, comparing with Propositions E.2, E.7, E.11 and E.12, the convergence rates have decreased to a certain extent, ultimately resulting in a larger regret.

### E.2.2   Proof of Theorem 3.1

*Proof.* Basically, similar to the proof of Theorem 3.2, we decompose the SubOpt$(\cdot)$ into two parts.

$$
\begin{aligned}
&\text{SubOpt}_t(a_t) \\
&\leq \mathbb{E}|a_t^* * Q^e(P_{St}, P_{Dt}, a_t^*) - a_t * Q^e(P_{St}, P_{Dt}, a_t)| + \\
&\quad\; \mathbb{E}|a_t * Q^e(P_{St}, P_{Dt}, a_t) - a_t * Q^e(P_{St}, P'_{Dt}, a_t)|.
\end{aligned}
$$

We know that the optimal service fee at round $t$ is $a_t^* = \frac{\beta_0 - \alpha_0}{2}$ while the service fee we set is $a_t = \frac{\widehat{\beta}_{0t} - \alpha_0}{2} + \epsilon_t$. Therefore, considering the Taylor's expansion, it holds that

$$
\mathbb{E}|a_t^* * Q^e(P_{St}, P_{Dt}, a_t^*) - a_t * Q^e(P_{St}, P_{Dt}, a_t)| \lesssim \mathcal{O}(\frac{\log \log T}{\sqrt{t}} + \frac{1}{t(1-\gamma)^2} + \epsilon_t^2),
$$

where the expectation is taken over $\epsilon_{Dt}$ due to Proposition E.15 and $(a+b+c)^2 \leq 3(a^2 + b^2 + c^2)$.

Considering the $m$-th episode, we know that there are $n = 2^m$ rounds in it. Since we already have that

$$
\frac{1}{n}|\sum_{i=1}^{n} \epsilon_i^2 - \sum_{i=1}^{n} \frac{1}{\sqrt{i}}| \lesssim \mathcal{O}(\frac{\sqrt{\log T \log \log T}}{n}),
$$

with probability at least $1 - \frac{\delta}{\log_2 T}$, it holds that

$$
\sum_{i=1}^{n} \epsilon_i^2 \lesssim \mathcal{O}(\sqrt{n} + \sqrt{\log T \log \log T}).
$$

This will lead to the following result that

$$
\sum_{t=1}^{T} \epsilon_t^2 \lesssim \mathcal{O}(\sqrt{T})
$$

because of the geometric series summation formula.

As a result, we get

$$\sum_{t=1}^{T} \mathbb{E}|a_t^* * Q^e(P_{St}, P_{Dt}, a_t^*) - a_t * Q^e(P_{St}, P_{Dt}, a_t)| \lesssim \mathcal{O}(\sqrt{T}\log\log T + \frac{\log T}{(1-\gamma)^2}),$$

with probability at least $1 - \delta$.

As for the second term $\mathbb{E}|a_t * Q^e(P_{St}, P_{Dt}, a_t) - a_t * Q^e(P_{St}, P'_{Dt}, a_t)|$, similarly, we have

$$\sum_{t=1}^{T} \mathbb{E}|a_t * Q^e(P_{St}, P_{Dt}, a_t) - a_t * Q^e(P_{St}, P'_{Dt}, a_t)| \leq \frac{\bar{P}\bar{Q}\sqrt{\gamma}\log_2 T}{1-\gamma}$$

due to Equation (2).

Recall that we also need that $n \gtrsim \mathcal{O}(\log T \log\log T + \frac{1}{(1-\gamma)^2})$ to make Proposition E.14 hold. Combining all these terms, it holds that

$$\begin{aligned}
\text{Regret}(T) &= \sum_{t=1}^{T} \text{SubOpt}_t(a_t) \\
&\lesssim \mathcal{O}(\sqrt{T}\log\log T + \frac{\log T}{(1-\gamma)^2}) + \frac{\bar{P}\bar{Q}\sqrt{\gamma}\log_2 T}{1-\gamma} + \mathcal{O}(\log T \log\log T + \frac{1}{(1-\gamma)^2}) \\
&\lesssim \mathcal{O}(\sqrt{T}\log\log T + \frac{\log T}{(1-\gamma)^2}),
\end{aligned}$$

with probability at least $1 - 10\delta$.

Recall that $\delta$ is exclusively concealed within $\log\log T$ terms. By assigning $\iota = 10\delta$, we have reached the following conclusion that

$$\text{Regret}(T) \lesssim \mathcal{O}(\sqrt{T}\log(\frac{\log T}{\iota}) + \frac{\log T}{(1-\gamma)^2}),$$

which finishes our proof. $\qquad\square$

# F Omitted Proof in Section 4

## F.1 Useful Facts for Proving Theorem 4.1

First, we introduce some auxiliary lemmas on Kullback–Leibler (KL) divergence.

**Lemma F.1.** *For two independent normal distributions $p = \mathcal{N}(\mu_1, \sigma_1^2)$ and $q = \mathcal{N}(\mu_2, \sigma_2^2)$, the KL divergence between them is*

$$KL(p\|q) = \frac{\sigma_1^2 + (\mu_1 - \mu_2)^2}{2\sigma_2^2} + \log(\frac{\sigma_2}{\sigma_1}) - \frac{1}{2}.$$

*Proof.* Recall that $KL(p\|q) = \int_x p(x)\log(\frac{p(x)}{q(x)})dx$. It leads to the result with the expression of normal distribution and some calculations. $\qquad\square$

**Lemma F.2** ([68]). *For two probability distributions $p$, $q$ over space $(\Omega, \mathcal{F})$, it holds that for any $A \in \mathcal{F}$*

$$p(A) + q(A^c) \geq \frac{1}{2}e^{-KL(p\|q)}.$$

## F.2 Proof of Theorem 4.1

*Proof.* We only prove the lower bound against myopic buyer which is the loosest one. Let's consider the following case. We assume the supply curve is $P_S = Q_S$ and the family of the demand curve parameterized by $\epsilon$ is $\{P_{\epsilon D} = 20 + 5\epsilon - (1 + \epsilon)Q_D + \epsilon_{\epsilon D}\}$ where $\epsilon_{\epsilon D} \sim \mathcal{N}(0, (\frac{2+\epsilon}{2})^2)$.

The system is depicted completely by $(a_t, P_{\epsilon t}^e)$, where $Q_{\epsilon t}^e = P_{\epsilon t}^e - a_t$. Recall that $a_t$ depends on history up to $t-1$, namely, $\mathcal{H}_{t=1} = (a_1, P_{\epsilon 1}^e, ..., a_{t-1}, P_{\epsilon(t-1)}^e)$. From Lemma E.1, we know that

$$\begin{cases} Q_\epsilon^e = \frac{20+5\epsilon-a+\epsilon_{\epsilon D}}{2+\epsilon}, \\ P_\epsilon^e = \frac{20+5\epsilon+(1+\epsilon)a+\epsilon_{\epsilon D}}{2+\epsilon}. \end{cases}$$

Then, we know that $P_\epsilon^e \sim \mathcal{N}(\frac{20+5\epsilon+(1+\epsilon)a}{2+\epsilon}, \frac{1}{4})$ due to the definition of $\epsilon_{\epsilon D}$. Moreover, the ex-ante optimal service fee is $a^* = \frac{20+5\epsilon}{2}$.

We now consider the situation when $\epsilon = 0$ and $\epsilon = T^{-\frac{1}{4}}$ and we use $\mathbb{P}_\epsilon$ to denote associated probability measure. With the help of Lemma F.1, it holds that

$$KL(\mathbb{P}_0\|\mathbb{P}_\epsilon) = \mathbb{E}_a[KL(\mathbb{P}_0(\cdot\,|\,a)\|\mathbb{P}_\epsilon(\cdot\,|\,a))\,|\,a] = \mathbb{E}_a[\frac{\epsilon^2(10-a)^2}{2(2+\epsilon)^2}],$$

where we use the law of iterated expectations.

During the whole $T$ round, we calculate the number of rounds with $|10 - a_t| \geq 5\epsilon$ denoted by $T_0$. With a slight abuse of notation, we use $[T_0]$ to denote the corresponding index set. Let's consider achieved regret with $T_0$ case by case.

On the one hand, when $T_0 \geq \frac{T}{2}$, it holds that one-round suboptimlality for $\epsilon = 0$ is at least $\text{SubOpt}_{0t}(a_t) = \frac{(10-a)^2}{2}$ when $t \in [T_0]$. Then, we have

$$\begin{aligned} \text{Regret}_0(T) + \text{Regret}_\epsilon(T) &\geq \text{Regret}_0(T) \\ &\geq \sum_{t \in [T_0]} \text{SubOpt}_{0t}(a_t) \\ &\geq T_0 \frac{25\epsilon^2}{2} \\ &\geq \frac{T}{2}\frac{25}{2\sqrt{T}} \\ &\gtrsim \Omega(\sqrt{T}). \end{aligned}$$

The first inequality holds due to the positivity of regret while the second inequality holds due to the positivity of $\text{SubOpt}(\cdot)$. The third inequality holds because when $t$ belongs to $[T_0]$, we have that $|10 - a| \geq 5\epsilon$ while the fourth inequality holds due to the assumption of $T_0$.

On the other hand, when $T_0 < \frac{T}{2}$, there exists more than $\frac{T}{2}$ rounds such that $|10 - a| \leq 5\epsilon$. In these rounds, denoted by $[T - T_0]$, it holds that $KL(\mathbb{P}_0\|\mathbb{P}_\epsilon) \leq \frac{25\epsilon^4}{2(2+\epsilon)^2} \leq \frac{25\epsilon^4}{18}$ as $\epsilon \leq 1$. The first inequality holds with no need to consider the distribution of $a$ because we use a union bound over all $a$. Then, the total $KL$ among these $T - T_0$ rounds is no larger than $(T - T_0)\frac{25\epsilon^4}{18} \leq \frac{25}{18} \lesssim \mathcal{O}(1)$ as $\epsilon = T^{-1/4}$ in leverage of properties of $KL$ divergence.

We define an event $A$ that in more than $\frac{T}{4}$ rounds, $a_t \geq \frac{40+5\epsilon}{4}$. Therefore, $A^c$ contains at least $\frac{T}{4}$ rounds that $a_t < \frac{40+5\epsilon}{4}$. Therefore, it holds that

$$\begin{aligned} \mathbb{E}_{[\cdot\,|\,T_0<\frac{T}{2}]}\left[\text{Regret}_0(T) + \text{Regret}_\epsilon(T)\right] &\geq \mathbb{E}_{[\cdot\,|\,T_0<\frac{T}{2}]}\left(\sum_{t\in[T-T_0]} \text{SubOpt}_{0t}(a_t) + \text{SubOpt}_{\epsilon t}(a_t)\right) \\ &\geq \mathbb{P}_0(A\,|\,T_0<\frac{T}{2})\frac{T}{4}\frac{25\epsilon^2}{32} + \mathbb{P}_\epsilon(A^c\,|\,T_0<\frac{T}{2})\frac{T}{4}\frac{25\epsilon^2}{16(2+\epsilon)} \\ &\geq \frac{25\epsilon^2 T}{192}(\mathbb{P}_0(A\,|\,T_0<\frac{T}{2}) + \mathbb{P}_\epsilon(A^c\,|\,T_0<\frac{T}{2})) \\ &\geq \frac{25\sqrt{T}}{192}\frac{1}{2}e^{-\frac{25}{18}} \\ &\gtrsim \Omega(\sqrt{T}). \end{aligned}$$

The first inequality holds due to the positivity of $\mathrm{SubOpt}(\cdot)$ while the second inequality holds because the optimal service fee is $\frac{20+5\epsilon}{2}$ and one-round loss is $\frac{[a-(20+5\epsilon)/2]^2}{2+\epsilon}$. The third inequality holds due to $\epsilon \leq 1$ while the fourth inequality holds due to Lemma F.2.

Therefore, it holds that by combining both two cases

$$
\begin{aligned}
&\max\{\mathbb{E}\mathrm{Regret}_0(T), \mathbb{E}\mathrm{Regret}_\epsilon(T)\} \\
&\geq \frac{1}{2}(\mathbb{E}\mathrm{Regret}_0(T) + \mathbb{E}\mathrm{Regret}_\epsilon(T)) \\
&\geq \min\{\mathbb{E}_{[\cdot|T_0\geq\frac{T}{2}]}\frac{\mathrm{Regret}_0(T) + \mathrm{Regret}_\epsilon(T)}{2}, \mathbb{E}_{[\cdot|T_0<\frac{T}{2}]}\frac{\mathrm{Regret}_0(T) + \mathrm{Regret}_\epsilon(T)}{2}\} \\
&\geq \frac{1}{2}\min\{\frac{25\sqrt{T}}{4}, \frac{25\sqrt{T}}{384}e^{-\frac{25}{18}}\} \\
&\gtrsim \Omega(\sqrt{T}),
\end{aligned}
$$

due to $\epsilon = T^{-1/4}$. Therefore, for any algorithm against $\epsilon = 0$ and $\epsilon = T^{-1/4}$, at least one regret is no smaller than $\Omega(\sqrt{T})$, which ends the proof. □

There are two key points when constructing such a hard-to-learn instance. When $\epsilon = 0$, the optimal service fee is 10 and the expected equilibrium is $(P^e, Q^e) = (15, 5)$. Then, we choose a special family of demand curves which are all through $(P^e, Q^e) = (15, 5)$ in expectation. Therefore, when the platform chooses $a = 10$, it cannot infer the value of $\epsilon$ which ends with large regret for non-zero $\epsilon$. However, if the platform sets service fee deviated from 10, it will suffer a high loss in the case when $\epsilon = 0$, ending with large total regret. This introduces an internal exploration-exploitation tradeoff leading to $\Omega(\sqrt{T})$ regret loss bound. However, when $\sigma_S^2 > 0$, even if the platform sets optimal service fee for $\epsilon = 0$, the noise $\epsilon_S$ plays the role in exploring the environment. The platform can utilize such information to do casual inference which heavily reduces the regret to $\widetilde{\mathcal{O}}(1)$. So, we answer the question that noise helps learning essentially.

Secondly, from a technical perspective, we adaptively adjust the noise variance in the demand. For the tuned hyperparameter $\epsilon = T^{-\frac{1}{4}}$ and baseline parameter $\epsilon = 0$, we shrink $\sigma_{\epsilon D}^2$ from $(\frac{2+\epsilon}{2})^2$ to 1. Since the randomness of demand introduces noise into the equilibrium retrospectively, the equilibrium price and quantity appear nebula-like. The shrinkage of variance results in greater overlap between the two nebulae, which complicates the learning process. Mathematically, it causes the equilibrium price to have the same variance regardless of the value of $\epsilon$, entailing smaller KL divergence between different choices of demand. To sum up, these two crucial aspects make up this delicate and non-trivial example.

### F.3   A Weaker Version of Theorem 4.2

Note that there is a gap between the proposed upper bounds and corresponding lower bounds when $\sigma_S^2 \lesssim \mathcal{O}(\frac{1}{\sqrt{T}})$, indicating essential hardness in this interval. Therefore, we first give the following lemma which is a little bit weaker than the original Theorem 4.2 but depicts the characteristics in this area. Here, we use a constructive proof and then we give a complete information-theoretic proof of Theorem 4.2 in Appendix F.4.

---

**Algorithm 6** Oracle-Variant Algorithm.

---

**Input:** $T, \sigma_S^2$.
**Initialization:** $H_0 \leftarrow \emptyset$
**for** $t = 1$ **to** $T$ **do**
    Generate noise: $\epsilon_{St} \sim \mathcal{N}(0, \sigma_S^2)$.
    Generate virtual supply curve: $P'_{St} = \alpha_0 + \alpha_1 Q + \epsilon_{St}$.
    Call $\mathtt{Oracle}(\cdot, \cdot)$: $\widetilde{a}_t \leftarrow \mathtt{Oracle}(H_{t-1}, P'_{St})$.
    Select action: $a_t = \widetilde{a}_t + \epsilon_{St}$.
    Observe equilibrium: $o_t = (P_t^e, Q_t^e)$ from Equation (1).
    Update history: $H_t \leftarrow H_{t-1} \cup \{o_t, \epsilon_{St}\}$.
**end for**

---

**Lemma F.3.** *It's impossible to find an algorithm such that there exists some $\sigma_S^2 \lesssim o(\frac{1}{\sqrt{T}})$ and the corresponding expectation of regret belongs to $o(\sqrt{T})$, namely, $\mathbb{E}\mathrm{Regret}(T) \lesssim o(\sqrt{T})$.*

*Proof.* We have proof by contradiction to demonstrate this lemma. We assume that when the supply randomness is $\sigma_S^2 \lesssim o(\frac{1}{\sqrt{T}})$, there exists an algorithm denoted by `Oracle` whose expected regret belongs to $o(\sqrt{T})$. We use $H_t$ to represent history until round $t$, namely, $H_t = \{a_\tau, P_\tau^e, Q_\tau^e\}_{\tau=1}^t$. Therefore, `Oracle`$(\cdot, \cdot)$ is a mapping from $H_{t-1} \times P_{St}$ to $\mathbb{R}$. Now, we are going to show that if existing such `Oracle`, we can find a variant of it called `Oracle-Variant` which achieves $o(\sqrt{T})$ without supply randomness, contradicting Theorem 4.1. We assume $\gamma = 0$ which is enough to construct a counterexample. We present the detail of `Oracle-Variant` in Algorithm 6. The input is the number of rounds $T$ and the supply randomness $\sigma_S^2 \lesssim o(\frac{1}{\sqrt{T}})$ associated with `Oracle`.

We use $\widetilde{a}_t^*$ to denote the optimal service fee when there exists $\epsilon_{St} \sim \mathcal{N}(0, \sigma_S^2)$ and $a_t^*$ to denote the one without randomness. Considering the following counterfactual case, if we adopt $\widetilde{a}_t$ when there exists $\epsilon_{St}$, the equilibrium price and quantity will be

$$(\widetilde{P_t^e}, \widetilde{Q_t^e}) = (\frac{\beta_0 - \alpha_0 - \widetilde{a}_t + \epsilon_{Dt} - \epsilon_{St}}{\alpha_1 - \beta_1}, \frac{\alpha_1\beta_0 - \alpha_0\beta_1 - \beta_1\widetilde{a}_t + \alpha_1\epsilon_{Dt} - \beta_1\epsilon_{St}}{\alpha_1 - \beta_1})$$

according to Lemma E.9. Fortunately, choosing $a_t = \widetilde{a}_t + \epsilon_{St}$ leads to the same equilibrium price and quantity without randomness, that is to say, $(P_t^e, Q_t^e) = (\widetilde{P_t^e}, \widetilde{Q_t^e})$.

Since we use `Oracle`, the regret when baseline is $\widetilde{a}_t^*$ is $o(\sqrt{T})$. Therefore, we know that

$$\mathbb{E}[\sum_{t=1}^T (\widetilde{a}_t - \widetilde{a}_t^*)^2] \lesssim o(\sqrt{T}),$$

according to the proof of Theorem E.1. Now, let's give a decomposition of $(a_t - a_t^*)^2$. It holds that

$$(a_t - a_t^*)^2 \lesssim \mathcal{O}((a_t - \widetilde{a}_t)^2) + \mathcal{O}((\widetilde{a}_t - \widetilde{a}_t^*)^2) + \mathcal{O}((a_t^* - \widetilde{a}_t^*)^2),$$

according to simple algebra. For the first term, we know that

$$\mathbb{E}[\sum_{t=1}^T \mathcal{O}((a_t - \widetilde{a}_t)^2)] \lesssim \mathbb{E}[\mathcal{O}(\sum_{t=1}^T \epsilon_{St}^2)] \lesssim \mathcal{O}(T\sigma_S^2),$$

from the implementation of Algorithm 6. For the second term, it holds that

$$\mathbb{E}[\sum_{t=1}^T \mathcal{O}((\widetilde{a}_t - \widetilde{a}_t^*)^2)] \lesssim o(\sqrt{T}).$$

For the last term, we know that

$$\mathbb{E}[\sum_{t=1}^T \mathcal{O}((a_t^* - \widetilde{a}_t^*)^2)] \lesssim \mathbb{E}[\sum_{t=1}^T \mathcal{O}(\epsilon_{St}^2)] \lesssim \mathcal{O}(T\sigma_S^2),$$

because $|a_t^* - \widetilde{a}_t^*| = |\frac{\epsilon_{St}}{2}|$ according to Lemma E.1.

Therefore, we know that without supply randomness, the expected regret of Algorithm 6 satisfying

$$\mathrm{Regret}(T) \lesssim o(\sqrt{T}) + \mathcal{O}(T\sigma_S^2) \lesssim o(\sqrt{T}).$$

The last inequality holds due to $\sigma_S^2 \lesssim o(\frac{1}{\sqrt{T}})$. However, from Theorem 4.1, we know that $\mathbb{E}\mathrm{Regret}(T) \gtrsim \Omega(\sqrt{T})$. It then causes a contradiction. We subsequently conclude that the existence of `Oracle` is untenable, which ends the proof. □

### F.4 Proof of Theorem 4.2

*Proof.* Let $\beta = (\beta_0, \beta_1)$ and $\epsilon_{Dt} \sim \mathcal{N}(0, \sigma_D^2)$ with $\sigma_D = (\alpha_1 - \beta_1)\sigma$ for some $\sigma > 0$. To express concisely, we use subscript $\beta$ and superscript $\pi$ to denote parameter and policy, respectively. We know that

$$a_t^*(\beta) = \frac{\beta_0 - \alpha_0 - \epsilon_{St}}{2}.$$

Also, when the decision maker sets any $a_t$, we observe

$$Q_t^e = \frac{\beta_0 + \epsilon_{Dt} - a_t - \alpha_0 - \epsilon_{St}}{\alpha_1 - \beta_1}$$

Note that the only unknown randomness comes from $\epsilon_{Dt}$ since $\epsilon_{St}$ is always observable. Meanwhile, upon deciding $a_t$ and observing $Q_t^e$ and $\epsilon_{St}$, the price from both the demand and supply side can be uniquely decided. Therefore, the log-likelihood prior to time $t$ can be calculated as:

$$\mathcal{L}_{t-1}(\beta) = \sum_{i=1}^{t-1} -\frac{1}{2\sigma^2}\left(Q_i^e - \frac{\beta_0 - a_i - \alpha_0 - \epsilon_{Si}}{\alpha_1 - \beta_1}\right)^2 + C,$$

where $C$ is a constant only dependent on $\{a_i, Q_i^e\}_{i=1}^{t-1}$ and *not* dependent on $\beta$. The fisher information matrix prior to time $t$ can be calculated as

$$\mathcal{F}_{t-1}^\pi(\beta) = \mathbb{E}_\beta^\pi\left[-\partial^2\mathcal{L}_{t-1}(\beta)/\partial\beta^2\right] = \frac{1}{\sigma^2}\mathbb{E}_\beta^\pi\left[\sum_{i=1}^{t-1}\left[\begin{matrix}\frac{1}{(\alpha_1-\beta_1)^2} & -\frac{\beta_0-a_t-\alpha_0-\epsilon_{St}}{(\alpha_1-\beta_1)^2} \\ -\frac{\beta_0-a_t-\alpha_0-\epsilon_{St}}{(\alpha_1-\beta_1)^2} & \frac{(\beta_0-a_t-\alpha_0-\epsilon_{St})^2}{(\alpha_1-\beta_1)^2}\end{matrix}\right]\right].$$

Let $\lambda$ be an absolutely continuous density on $\Theta$, taking positive values on the interior of $\Theta$ and zero on its boundary (see, e.g., Keskin and Zeevi [44]). Then the multivariate Van Trees inequality [36] implies that

$$\mathbb{E}_\lambda\left[\mathbb{E}_\beta^\pi\left[(a_t - a_t^*(\beta))^2\right]\right] \geq \frac{\left(\mathbb{E}_\lambda\left[C(\beta)(\partial a_t^*(\beta)/\partial\beta)^\top\right]\right)^2}{\mathbb{E}_\lambda\left[C(\beta)\mathcal{F}_{t-1}^\pi(\beta)C(\beta)^\top\right] + \widetilde{\mathcal{F}}(\lambda)}. \tag{3}$$

Let $C(\beta) = [(\beta_0 - \alpha_0)/2 \quad 1]$. Then we have

$$C(\beta)(\partial a_t^*(\beta)/\partial\beta)^\top = [(\beta_0-\alpha_0)/2 \quad 1][1/2 \quad 0]^\top = (\beta_0-\alpha_0)/4 = \Omega(1)$$

and

$$C(\beta)\mathcal{F}_{t-1}^\pi(\beta)C(\beta)^\top = \frac{1}{\sigma^2(\alpha_1-\beta_1)^2}\mathbb{E}_\beta^\pi\left[\sum_{i=1}^{t-1}\left(\frac{\beta_0-\alpha_0}{2} - (\beta_0 - a_t - \alpha_0 - \epsilon_{St})\right)^2\right]$$

$$= \frac{1}{\sigma^2(\alpha_1-\beta_1)^2}\sum_{i=1}^{t-1}\mathbb{E}_\beta^\pi\left[\left(a_i - \frac{\beta_0-\alpha_0-\epsilon_{Si}}{2} + \frac{\epsilon_{Si}}{2}\right)^2\right]$$

$$= \mathcal{O}\left(\sum_{i=1}^{t-1}\left(\mathbb{E}_\beta^\pi\left[(a_i - a_i^*(\beta))^2\right] + \sigma_S^2\right)\right).$$

Plugging the inequalities above into Inequality (3), we know that there exists a positive constant $c$ such that

$$\mathbb{E}_\lambda\left[\mathbb{E}_\beta^\pi\left[(a_t - a_t^*(\beta))^2\right]\right] \geq \frac{2c}{1 + \sum_{i=1}^{t-1}\mathbb{E}_\lambda\left[\mathbb{E}_\beta^\pi\left[(a_i - a_i^*(\beta))^2\right]\right] + (t-1)\sigma_S^2}.$$

From now on, for notation simplicity, we denote $\Delta^{[t,T]} = \sum_{i=t}^T\mathbb{E}_\lambda\left[\mathbb{E}_\beta^\pi\left[(a_i - a_i^*(\beta))^2\right]\right]$. Summing the formula above from $t$ to $T$, we know that

$$\Delta^{[t,T]} \geq \sum_{i=t-1}^{T-1}\frac{2c}{1 + \Delta^{[1,i]} + \sigma_S^2 \cdot i}. \tag{4}$$

Then $\sup_\beta \text{Regret}_\beta^\pi(T) = \Theta(\Delta^{[1,T]})$. We show the lower bound in two cases.

(a). $T \leq 1 + 1/\sigma_S^4$. Then from Inequality (4) we have

$$\Delta^{[1,T]} \geq \sum_{i=1}^{T-1}\frac{2c}{1 + \Delta^{[1,i]} + \sigma_S^2 \cdot i} \geq \frac{2c(T-1)}{1 + \Delta^{[1,T]} + \sqrt{T}} \geq \frac{cT}{\Delta^{[1,T]} + 2\sqrt{T}}.$$

Therefore,

$$(\Delta^{[1,T]} + \sqrt{T})^2 \geq (1+c)T,$$

which indicates that there exists a constant $c_0 = \sqrt{1+c} - 1 > 0$ such that $\Delta^{[1,T]} \geq c_0\sqrt{T}$.

(b). $T > 1 + 1/\sigma_S^4$. Let $K$ be the smallest positive integer such that $2^K \geq 1 + 1/\sigma_S^4$. Then $2^{K-1} < 1 + 1/\sigma_S^4$, which indicate that

$$K + 1 = K - 1 + 2 < \log_2(1 + 1/\sigma_S^4) + 2 \leq 6(1 + \log_+(1/\sigma_S)).$$

From (a) we know that for $t = 2^K$, we have

$$\Delta^{[1,t]} \geq c_0\sqrt{1 + 1/\sigma_S^4} \geq c_0 \frac{1}{\sigma_S^2} \geq \frac{c_1}{1 + \log_+(1/\sigma_S)} \frac{K}{\sigma_S^2},$$

where $c_1$ is a small positive constant irrelevant with $K$ and $\sigma_S$ such that $c_1(2 + 6c_1) \leq c$.

Now we use induction. Suppose we have

$$\Delta^{[1,2^k]} \geq \frac{c_1}{1 + \log_+(1/\sigma_S)} \frac{k}{\sigma_S^2} \tag{5}$$

for some $k \geq K$. From Inequality (4) we have

$$\Delta^{[2^k+1, 2^{k+1}]} \geq \frac{(2^{k+1} - 2)c}{1 + \Delta^{[1,2^{k+1}]} + \sigma_S^2 \cdot (2^{k+1} - 1)} \geq \frac{c}{2^{-k} + \frac{\Delta^{[1,2^{k+1}]}}{2^k} + \frac{3}{2}\sigma_S^2} \geq \frac{c}{\frac{\Delta^{[1,2^{k+1}]}}{2^k} + 2\sigma_S^2}.$$

Note that we have utilized the following inequality:

$$2^{-k} \leq \frac{\sigma_S^4}{1 + \sigma_S^4} \leq \frac{\sigma_S^4}{2\sigma_S^2} = \sigma_S^2/2.$$

As a result,

$$\Delta^{[1,2^{k+1}]} \geq \frac{c_1}{1 + \log_+(1/\sigma_S)} \frac{k}{\sigma_S^2} + \frac{c}{\frac{\Delta^{[1,2^{k+1}]}}{2^k} + 2\sigma_S^2}.$$

If $\Delta^{[1,2^{k+1}]} < \frac{c_1}{1 + \log_+(1/\sigma_S)} \frac{k+1}{\sigma_S^2}$, then

$$\frac{c}{\frac{\Delta^{[1,2^{k+1}]}}{2^k} + 2\sigma_S^2} > \frac{c}{\frac{c_1}{1 + \log_+(1/\sigma_S)} \frac{k+1}{2^k\sigma_S^2} + 2\sigma_S^2}$$

$$\geq \frac{c}{\frac{c_1}{1 + \log_+(1/\sigma_S)} \frac{K+1}{2^K\sigma_S^2} + 2\sigma_S^2}$$

$$\geq \frac{c}{(2 + 6c_1)\sigma_S^2}$$

$$\geq \frac{c_1}{\sigma_S^2}.$$

This causes a contradiction. Therefore, for any $k \geq K$, Inequality (5) holds. Now select any $T > 1 + 1/\sigma_S^4$. If $T < 2^K$, then

$$\Delta^{[1,T]} \geq c_0\sqrt{1 + 1/\sigma_S^4} \geq \frac{c_1}{1 + \log_+(1/\sigma_S)} \frac{K}{\sigma_S^2} \geq \frac{c_1}{1 + \log_+(1/\sigma_S)} \frac{\log T}{\sigma_S^2}.$$

If $T \geq 2^K$, choose $k = \lfloor \log_2 T \rfloor$, then

$$\Delta^{[1,T]} \geq \Delta^{[1,2^k]} \geq \frac{c_1}{1 + \log_+(1/\sigma_S)} \frac{k}{\sigma_S^2} \geq \frac{c_1/2}{1 + \log_+(1/\sigma_S)} \frac{\log T}{\sigma_S^2}.$$

$\square$

### F.5 Proof of Theorem 4.3

*Proof.* Note that the buyer's strategic behavior only causes extra $\mathcal{O}(\frac{\log T}{\sigma_S^2})$ in Theorem 3.2 and $\mathcal{O}(\log T)$ in Theorem 3.1. Since terms from estimation errors dominate them, we without loss of generality assume that the buyer is myopic i.e., $\gamma = 0$ for simplicity without weakening our results.

From traditional results of hypothesis test [22], we know we need $n \asymp \Theta((\frac{z_\alpha \sigma}{E})^2)$ samples to distinguish $\mathbb{H}_0$ and $\mathbb{H}_1$, where $z_\alpha$, $\sigma$ and $E$ are the $z$-score, the population standard deviation and the acceptable margin of error, respectively. Here, we know that $\sigma \asymp \Theta(\sigma_S^2)$ since $\mathrm{Var}(\epsilon_{St}^2) \asymp \Theta(\sigma_S^4)$. Besides, we set $\alpha \asymp \Theta(\frac{1}{T})$ and then $z_\alpha \asymp \Theta(\sqrt{\log T})$ due to Lemmas D.1 and D.2. The largest tolerance of error is $E \asymp \Theta(\frac{1}{\sqrt{T}})$. Therefore, by setting $T_0 \asymp \Theta(\log T)$ with some accurately designed constants, we can distinguish $\mathbb{H}_0$ and $\mathbb{H}_1$ with probability at least $1 - \Theta(\frac{1}{T})$.

Then, in the case of $\mathbb{H}_0$, we will add some human-made noise referring to Algorithm 5. Then, the randomness of $a_t$ comes from two parts. One is from original $\epsilon_{St}$. The other one comes from artificially added noise. Therefore, the variance of $a_t$ is no smaller than the one in the proof of Theorem 3.1 and the estimation of parameters will be more accurate. Subsequently, we can bound three parts of regret which construct the total $\mathrm{Regret}(T)$. The first one comes from the gap between $a_t$ and $a_t^*$. It will lead to an $\widetilde{\mathcal{O}}(\sqrt{T})$ regret. The second part comes from the fact that the probability that some inequalities don't hold is at most $\mathcal{O}(\frac{1}{T})$. The last part comes from the first $T_0$ rounds. Therefore, it holds that

$$\mathbb{E}\mathrm{Regret}(T) \lesssim \widetilde{\mathcal{O}}(\sqrt{T}) + T * \mathcal{O}(\frac{1}{T}) + \mathcal{O}(T_0) \lesssim \widetilde{\mathcal{O}}(\sqrt{T}).$$

Recall that we will suffer st most extra $\mathcal{O}(\log T)$ from a strategic buyer which is indeed a subdominant term. It covers cases when $\sigma_S^2 \lesssim o(\frac{1}{\sqrt{T}})$.

In the case of $\mathbb{H}_1$, it will reduce to Theorem E.1 directly. The regret consists of three parts as well. The first part is $\widetilde{\mathcal{O}}(\frac{1}{\sigma_S^2})$ inherited from Theorem E.1. The second term comes from the $\mathcal{O}(\frac{1}{T})$ probability of some inequalities being violated. The last term is from the first $T_0$ rounds, yielding

$$\mathbb{E}\mathrm{Regret}(T) \lesssim \widetilde{\mathcal{O}}(\frac{1}{\sigma_S^2}) + T * \mathcal{O}(\frac{1}{T}) + \mathcal{O}(T_0) \lesssim \widetilde{\mathcal{O}}(\frac{1}{\sigma_S^2}).$$

Note that we will suffer an extra $\mathcal{O}(\frac{\log T}{\sigma_S^2})$ regret facing a far-sighted buyer. This case contains all situations when $\sigma_S^2 \gtrsim \omega(\frac{1}{\sqrt{T}})$. To be specific, from the proof of Theorems 3.1 and 3.2, we know we hide $\mathcal{O}(\log^2 T)$ for $\mathbb{H}_1$ but only $\mathcal{O}(\log T)$ for $\mathbb{H}_0$.

Finally, when $\sigma_S^2 \asymp \Theta(\frac{1}{\sqrt{T}})$, the policy we use might be a mixture of the one under $\mathbb{H}_0$ and the one under $\mathbb{H}_1$. Luckily, both of them will lead to an at most $\widetilde{\mathcal{O}}(\sqrt{T})$ expected regret.

To sum up, we know that the expected regret of Algorithm 3 satisfies

$$\mathbb{E}\mathrm{Regret}(T) \lesssim \begin{cases} \widetilde{\mathcal{O}}(\sqrt{T}) & \text{when } \sigma_S^2 \lesssim \mathcal{O}(\frac{1}{\sqrt{T}}) \\ \widetilde{\mathcal{O}}(\frac{1}{\sigma_S^2}) & \text{when } \sigma_S^2 \gtrsim \Omega(\frac{1}{\sqrt{T}}). \end{cases}$$

Let's now turn to the tightness of lower bounds. From Theorems 4.1 and 4.2, we have that

$$\mathbb{E}\mathrm{Regret}(T) \gtrsim \begin{cases} \Omega(\sqrt{T}) & \text{when } \sigma_S^2 = 0 \\ \Omega(\sqrt{T}) & \text{when } \sigma_S^2 \lesssim \mathcal{O}(\frac{1}{\sqrt{T}}) \\ \Omega(\frac{1}{\sigma_S^2}) & \text{when } \sigma_S^2 \gtrsim \Omega(\frac{1}{\sqrt{T}}). \end{cases}$$

Therefore, we conclude that our regret lower bounds are tight and Algorithm 3 is optimal regardless of constants and logarithmic terms.

Moreover, we instantly obtain the phase transition of regret and it finishes our proof. $\square$

### F.6 A Lower Bound When $\gamma = 1$

We close off this section by giving an easy reduction to the $\Omega(T)$ lower bound in Amin et al. [2] when the buyer is patient, i.e., equipped with discount rate 1. To determine the optimal dependency on $\gamma$ is s beyond the scope of the current work and we leave it as an interesting future direction.

For each $\beta_0$, we have a unique optimal service fee $a^*$ corresponding to it. In the meanwhile, $a^*$ is the marginal willingness to pay and we focus on the pricing problem for those with value $a^*$. From Theorem 3 in Amin et al. [2], we know there exists a value $v$, i.e., $a^*$ here, an $\Omega(T)$ regret is inevitable for any algorithm when the buyer is patient, namely, $\gamma = 1$. Through the bijection between $\beta_0$ and $a^*$, we know there exists some $\beta_0$ that makes this pricing problem unlearnable.

# G    Details of Numerical Experiments

We choose parameters $\gamma = 0$, $\alpha_0 = 1$, $\alpha_1 = 1$, $\beta_0 = 5$, $\beta_1 = -1$ and $\epsilon_D \sim \mathcal{N}(0, 1)$ in both experiments.

In Section 5, we notice that the increment of regret has several stages. At the beginning of each stage, the regret will increase relatively rapidly, and then the rate decrease gradually. The reason behind this phenomenon is the way we implement $\texttt{Act}(\cdot, \cdot)$. We actually update our pricing policy at the start of each stage. Therefore, the noise we add has a variance that starts from 1 and decreases according to the inverse square root law.

For the second experiment, there are some hyper-parameters when implementing Algorithm 3. First, we set $T_0 = 10 \log_2 T$ therein. Secondly, when we do the hypothesis test, we set $\mathbb{H} = \mathbb{H}_0$ if the sample mean of $\epsilon_S^2$ is smaller than $\frac{10}{\sqrt{T}}$ and $\mathbb{H}_1$ otherwise. Among the 1000 choices of $\sigma_S^2$, there are about 100 cases where $\mathbb{H} = \mathbb{H}_0$, nearly to say, when $\sigma_S^2 \leq 0.1$. These choices of hyper-parameters are not essential and not well-tuned though enough for our experiments. When using LOWESS, we set the hyperparameter "fraction" as 0.15. It means that we use 15% nearest points when fitting the needed function value. Similarly, it is not carefully selected but rather folklore. Besides, note that $\log T \approx 9.21$ so regret fluctuating between 300 and 600 around $\sigma_S^2 = 0.1$ should be acceptable.

In conclusion, the first experiment testifies regret upper bounds in Theorems 3.1 and 3.2 while the second experiment validates results about the tightness of our lower bounds and the existence of phase transition in Theorem 4.3 in a concrete manner.

