# OpenReview forum: "Dynamic Service Fee Pricing under Strategic Behavior: Actions as Instruments and Phase Transition"
_NeurIPS.cc/2024/Conference — NeurIPS 2024 poster_

### Official Review · Reviewer_pXqU · 2024-07-12

**Soundness:** 3
**Presentation:** 2
**Contribution:** 2
**Rating:** 7
**Confidence:** 2

**Summary:**

The paper studies the problem of dynamic pricing of service fees in a setting where only equilibrium quantities of supply and demand curves are observable. The main contributions of the paper lies in using Instrumental Variables in an online setting, and consequently, theoretically bounding the regret of the model.

**Strengths:**

The problem is very interesting and the modelling is very realistic than hypothetical.
Theoretical Analysis.

**Weaknesses:**

Some proofs could had been included in the main paper as theoretical contribution is the strength of the paper.

**Questions:**

The equilibrium equation (1) between supply and demand was unclear. How does the equation be helpful for the buyer’s strategies. Some intuition could be provided here

**Limitations:**

The paper makes good theoretical observations about the setting in consideration. However, the 100% civilian-run seller assumption could be relaxed to make it a more impactful contribution. Other limitations are stated by the authors and these are not straightforward to fix and I agree with authors that these should be left for future.

---

> ### Author Rebuttal · Authors · 2024-08-05
>
> Thank you for your comments and encouragement. The followings are our responses to your questions.
>
> Re Weaknesses: Thank you for your suggestion. In the camera-ready version, we will include some core proofs on the additional page of the main body.
>
> Re Questions: $P_{St}$ represents the payment received by the seller, reflecting the supply curve, while $P_{Dt}$ represents the price paid by the buyer, reflecting the demand curve. Since the platform charges a service fee $a_t$, there is a gap between these two prices at equilibrium. In other words, at the equilibrium point where we use $Q_t^e$ to denote equilibrium quantity, we have $P_{Dt}(Q_t^e) = P_{St}(Q_t^e) + a_t$. However, buyers might not purchase according to their true demand, i.e., their willingness to pay. For example, a buyer may be able to afford Uber's price but choose to wait for some time until the price drops. Consequently, Uber may perceive the buyer's purchasing power as lower and reduce prices in future personalized pricing, thereby giving the buyer higher long-term utility. As a result, we use $P_{Dt}'$ to represent the buyer's apparent demand, based on which they choose the quantity to purchase, and it might be different from $P_{Dt}$ reflecting strategic behaviors. Therefore, we finally have $P_{Dt}'(Q_t^e) = P_{St}(Q_t^e) + a_t$.
>
> Re Limitations: Thank you for your comment. In Appendix B, we discuss how to generalize the 100% civilian-run seller assumption to $\alpha$ civilian-run. We chose $\alpha=100$% to concisely express our main contributions, namely (1) how to use endogenous actions as IVs, (2) how to deal with buyers' strategic behavior, and (3) how to explore the unknown environment through adaptive randomness injection.
>
> Thank you again for your comments. We hope you find the response satisfactory. Please let us know if you have further questions.

---

> > ### Comment · Reviewer_pXqU · 2024-08-12
> >
> > Thanks for the response. I retain my score.

---

> > > ### Author Response · Authors · 2024-08-14
> > >
> > > Thanks a lot!

---

### Official Review · Reviewer_EYLY · 2024-07-13

**Soundness:** 3
**Presentation:** 4
**Contribution:** 3
**Rating:** 6
**Confidence:** 4

**Summary:**

This paper considers a dynamic pricing problem where the buyer is strategic and provides regret analysis under this setting. In particular, they propose to use instrumental variables to estimate demand and discuss the impact of the supply randomness.

**Strengths:**

1. The paper is technically strong with several results and the presentation is clear: the study of "noise helps learning" and phase transition is  interesting
1. The setting of supply equilibrium is nonstandard in dynamic pricing and less explored

**Weaknesses:**

1. The problem motivation is questionable as the service fee does not change often in practice whereas the online learning problem considered effectively relies on frequent changes (equivalently, a large number of periods)
1. It is a pity that the regret dependence on $\gamma$ is not captured in the current analysis, as also mentioned in the future direction of this paper.

**Questions:**

1. As mentioned in Example 1.1,  the service fee is constant in a short period of time, and might not be that dynamic (compared to typical applications of dynamic pricing such as retailing or ride-hailing pricing). However, the numerical experiments involve a very large number fo time periods, in particular, the regret converges at around 20,000. The inconsistencies make the proposed algorithm inviable in practice. Can the authors reconcile the inconsistency and provide a more convincing motivation?
1. Can the authors explain a bit more on the revelation principle and in particular how the buyer maximizes surplus?
1. The paper mentions using non-i.i.d. action as an instrument; however, a rigorous exposition on conditional independence (i.e., sanity check) is missing, which decreases the theoretical credibility of this work. Can the authors enrich this part?
1. The noise-agnostic algorithm in Algorithm 3 is interesting but the current description is a bit concise. Can authors add explanation on how to implement the hypothesis test?
1. The confidence region is rather wide in Figure 2. Can authors check the algorithm performances and maybe present more results with different $\sigma_S$ values? It also might be worthwhile to increase the number of trajectories from $10$.

---

> ### Author Rebuttal · Authors · 2024-08-05
>
> Thank you for your detailed remarks and questions. The followings are our responses to your questions.
>
> Re Weakness 1 and Question 1: One application scenario could be that some online platforms, e.g., Amazon, sells a large number of products daily, and takes service fees between sellers and buyers.
> At the same time, we notice that on ride-hailing platforms like Uber, prices in the app frequently fluctuate due to frequent changes in supply. While the booking fee doesn't change too often, promotions can change quickly and be highly personalized. Therefore, our model can also be applied to dynamic promotion pushes. This is another application scenario for our model.
>
> Re Weakness 2: We agree that this is an important and challenging future direction. We would like to note that although our results on $\gamma$ may not be tight, we have improved upon previous results in the literature. Please refer to Lines 245-252. Compared to $O(1/(1-\gamma)^2)$ regret in another similar pricing problem [37], we improved it to $O(1/(1-\gamma))$ in Thm 3.2. Additionally, our algorithm does not require prior knowledge about $\gamma$.
>
> Re Question 2: The revelation principle [63] tells us that for any mechanism that maximizes revenue, there exists an incentive-compatible (IC) mechanism that can achieve the same revenue. IC means that buyers can purchase according to their true demand, and the mechanism's incentives will guarantee them higher utility. Therefore, we study how to motivate and approximate within the IC mechanism framework, narrowing down the policy space and making it easier to find an approximately optimal mechanism. This not only simplifies the analysis but also makes it easier to apply in real-world scenarios (buyers don't need to struggle with searching for a good strategy).
>
> Re Question 3: $a_t$ is not i.i.d. because it depends on the supply from $1$ to $t$ and the demand from $1$ to $t-1$. Therefore, $a_m$ and $a_n$ are correlated, since they, for example, both depend on the supply at $t=1$. However, since the platform announces $a_t$ before observing the equilibrium at time $t$, $a_t$ and $P_{Dt}$ are conditionally independent given $P_{S1}, ..., P_{St}, P_{D1}, ..., P_{D(t-1)}$. Therefore, we model it as a martingale for parameter estimation. Thank you for your comment; we will include this discussion in the next version.
>
> Re Question 4: In our experiments, we chose $T_0 = 10\log_2 T$. We also used $T_0$ data points to empirically estimate $\sigma_S^2$. If the estimated value is greater than $10/\sqrt{T}$, we consider it to belong to $\mathbb{H}_1$; otherwise, we consider it to belong to $\mathbb{H}_0$. The parameter 10 does not affect the order of the regret bound in terms of $T$, and the concrete value is usually determined empirically, such as through cross-validation with past data. We hope this helps with your understanding.
>
> Re Question 5: Thank you for your suggestion. We increased the number of trajectories to 100 and observed that the bandwidth significantly decreased. Please refer to Figure 1 in the newly added pdf. Additionally, we tested the regret under different $\sigma_S^2$ values: 0.5, 1, 1.5, 2. Please refer to Figures 2-4 in the newly added pdf. We found that the larger the $\sigma_S^2$, the smaller the regret, which confirms our theoretical results.
>
> Thank you again for your comments. We hope you find the response satisfactory. Please let us know if you have further questions.

---

> > ### Comment · Reviewer_EYLY · 2024-08-12
> > **Score adjusted**
> >
> > Thank you for your thoughtful feedback. Overall this is a technically strong and well-rounded paper. My previous concerns are largely well addressed and I appreciate the authors' efforts in addressing them. I have thus adjusted my score accordingly. Last minor note, though I remain a little skeptical about the motivation. As the authors have replied "While the booking fee doesn't change too often, promotions can change quickly and be highly personalized. Therefore, our model can also be applied to dynamic promotion pushes." -- if this is the case, it is probably more appropriate not to use service pricing as the problem background (just for the sake of being unique) if it does not fit the model well.

---

> > > ### Author Response · Authors · 2024-08-12
> > >
> > > Thank you for your feedback! Our analysis of the model is twofold. First, charging service fees is critical for third-party platforms as it constitutes a significant portion of a company's revenue. This work makes initial attempts to integrate various factors of service fee pricing into an intuitive model, illustrating the concept of actions as instruments and the phenomenon of phase transition. Second, our paper adopts an experimental approach, commonly used by many online platforms. If a platform lacks prior market information, it must explore to learn about the environment while maximizing total rewards. This includes dynamically adjusting service fees and personalizing promotional efforts, both fitting well within such an experimental framework. Nervertheless, we look forward to incorporating promotions and developing more practical models as an exciting avenue for future research. Please let us know if you have any more questions. Thank you once again for your insights!

---

### Official Review · Reviewer_Z4TZ · 2024-07-15

**Soundness:** 3
**Presentation:** 3
**Contribution:** 2
**Rating:** 5
**Confidence:** 4

**Summary:**

The primary goal of the paper is to maximize total revenue over a given time horizon by dynamically adjusting service fees on third-party platforms, considering the strategic nature of customers and the lack of complete information on demand.
The paper presents a novel approach to dynamic pricing for third-party platforms, addressing the challenges of unknown demand, equilibrium observation limitations, and strategic buyer behavior. The authors propose an algorithmic solution that incorporates active randomness injection, non-i.i.d. actions as instrumental variables, and a low-switching cost design to balance exploration and exploitation while estimating demand and mitigating strategic behavior.

**Strengths:**

1 The paper is technically rigorous, providing a clear mathematical framework and novel algorithmic solutions for a complex problem in platform economics.
2 The authors demonstrate an innovative use of non-i.i.d. actions as instrumental variables and the proposed algorithms are shown to have optimal regret bounds, which is a strong technical result.
3 The paper is well-structured, with a clear explanation of the problem, related work, model assumptions, and a detailed description of the algorithms and their theoretical guarantees.

**Weaknesses:**

1 While the paper is comprehensive, it could benefit from more real-world examples or case studies to illustrate the practical application of the proposed algorithms.


2 The assumptions made for the model may not always hold in real-world scenarios. For example, in reality, there is often a situation of diminishing marginal utility. so the assumption that supply and demand curves have linear forms hinder the practical application of this algorithm. The authors could discuss the robustness of the algorithms under different conditions.

3 The author say that they can consider incentive-compatible direct mechanism with the help of revelation principle. But the design space of the platform must be constrained compared to the original space. The authors do not discuss the impact of this change on their design of the algorithms.

**Questions:**

Could the authors provide more examples or case studies to illustrate how their algorithms perform in real-world scenarios?
How do the algorithms perform under different levels of noise in the supply curve, and what is their robustness to model misspecification?
The direct and IC mechanism could be constrained compared to the original space. Could the authors discuss the impact of this on their design of the algorithms?

**Limitations:**

I did not find any limitations of the paper.

---

> ### Author Rebuttal · Authors · 2024-08-05
>
> Thank you for your kind remarks and questions. The followings are our responses to your questions.
>
> Re Weakness 1: Thank you for your suggestion. If there is a suitable opportunity, we will consider conducting experiments on real-world data.
>
> Re Weakness 2: We are sorry for the confusion and misunderstanding. We discussed in Lines 165-177 why we use a linear model (based on market observations and the robustness of the linear model [15]) and how to go beyond it (using GMM and RKHS). We assumed a linear model to avoid heavy notations while expressing our main points: (1) how to use endogenous actions as IVs, (2) how to deal with buyers' strategic behavior, and (3) how to explore the unknown environment through adaptive randomness injection.
>
> Re Weakness 3 and Question 3: Recall that our goal is to maximize the total revenue. The revelation principle [63] tells us that there exists a DSIC mechanism that can achieve the optimal revenue. Therefore, we only need to consider the policy class satisfying DSIC. If we approach the optimal DSIC mechanism, then we can approach any optimal mechanism no matter it is DSIC or not. In other words, we identify there exists an optimal point in a smaller policy space and then approximate it within this smaller space. This not only simplifies the analysis but also makes it easier to apply in real-world scenarios (buyers don't have to struggle with searching for a good strategy).
>
> Re Question 1: Thanks for your question. One application scenario could be that some online platforms, e.g., Amazon, sells a large number of products daily, and takes service fees between sellers and buyers. At the same time, we notice that on ride-hailing platforms like Uber, prices in the app frequently fluctuate due to frequent changes in supply. While the booking fee doesn't change too often, promotions (which can be regarded as a reduction on the service fee) can change quickly and be highly personalized. Therefore, our model can also be applied to dynamic promotion pushes. This is another application scenario for our model.
>
> Re Question 2: We demonstrate the relationship between regret and the noise level in the supply curve in Figure 5. We also add another group of experiments (see pdf) showing their negative correlation. Additionally, a higher noise level in demand leads to greater regret, as it results in larger estimation errors for the parameters. Please refer to Line 688 for this impact on $\hat\beta_1$. [56] informs us that model misspecification causes additional unavoidable errors, while [15] explains that linear models are not so affected by misspecification. This is one of the reasons why we assume linearity.
>
> Thank you again for your comments. We hope you find the response satisfactory. Please let us know if you have further questions.

---

> > ### Comment · Reviewer_Z4TZ · 2024-08-13
> >
> > Thank you for the response. I understand that using a linear model simplifies notation and presentation. However, This also simplifies the learning process. It is still difficult to evaluate the performance of the algorithm in much more complicated, real-world applications. I will maintain my score.

---

> > > ### Author Response · Authors · 2024-08-13
> > >
> > > Thank you for your comment. We would like to emphasize that in this paper we take the first steps towards understanding service fee pricing in third-party platforms. We take an "actions as instruments" perspective and observe a "phase transition" phenomenon. We hope to extend the techniques to nonlinear functions in our next step (we believe the techniques and insights still hold). Thanks again for your review.

---

### Official Review · Reviewer_pz8n · 2024-07-23

**Soundness:** 3
**Presentation:** 3
**Contribution:** 2
**Rating:** 5
**Confidence:** 2

**Summary:**

The paper tackles the dynamic pricing problem faced by third-party platforms specifically concerning the optimal setting of service fees in the presence of strategic and far-sighted customers. The objective is to maximize the total revenue over a given time horizon.

**Strengths:**

+ The motivation and challenge of solving demand prediction and optimizing long-term revenue is clear.
+ Robustness consideration: the algorithm's design incorporates robustness by not requiring precise knowledge of the buyer's discount rate, enhancing its applicability across varying real-world scenarios.

**Weaknesses:**

-. Declaration: line 100, confusing definition of price P and demand Q.
-. Limitation: lack of explanation of assumption 1, regarding the linearity with Gaussian noise.
-. There is a lack of details on instrumental variables in Algorithm 1, which is crucial in solving the regret bounds since instrumental variables are non-iid and non-external.
-. The trade-offs of data amount and randomness are straightforward in Section 3.1.
-. Lack of motivation and explanation of relaxing the two assumptions in Section 3.2.
-. Lack of detailed definition of T_0 in Algorithm 3. The intuition of using hypothesis testing is quite straightforward.
-. In Section 5, to show the regret order, it’s better to show a log-like figure which indicates the linearity relation.

**Questions:**

see weakness.

**Limitations:**

NA.

---

> ### Author Rebuttal · Authors · 2024-08-05
>
> Thank you for your remarks. The followings are our responses to your questions.
>
> Re Weakness 1: $P_{St}$ is the amount received by the seller at time $t$, and $P_{Dt}$ is the amount paid by the buyer. The gap between them is the service fee, which is $a_t$, and $Q$ is the transaction quantity. Then, $P_{St}(Q)$ and $P_{Dt}(Q)$ represent the supply curve and demand curve respectively.
>
> Re Weakness 2: We are sorry for the confusion and misunderstanding. In Lines 165-177, we introduced the motivation and extension of Assumption 1. Our results hold for general subGaussian noise, and we use Gaussian noise for ease of illustration. We assume linear models because they perform well locally and have strong robustness.
>
> Re Weakness 3: We use the service fee, i.e., action, as an instrumental variable, which is emphasized in the title. Since they are neither i.i.d. nor external, we used a martingale-based approach. Intuitively, at time $t$, $a_t$ only depends on the demand from times 1 to $t-1$; so given those, $a_t$ is independent of the demand at time $t$. We proved that in this case, the service fee can be used as an approximately valid IV.
>
> Re Weakness 4: The challenge in our problem lies in how to add randomness. We carefully choose the amount of data used to make $a_t$ an approximately valid IV. Additionally, we add noise with carefully chosen magnitude to balance between exploration and exploitation.
>
> Re Weakness 5: For Assumption 2.1, we can use the generalized method of moments (GMM), which is standard in econometrics, to handle non-linearity. For Assumption 3.2, we discuss other possible choices in Appendix B. Please refer to the discussion in lines 154-177. We use these assumptions to highlight our technical innovations while avoiding heavy notations, including (1) how to use endogenous actions as IVs, (2) how to deal with buyers' strategic behavior, and (3) how to explore the unknown environment through adaptive randomness injection.
>
> Re Weakness 6: $T_0$ is a constant multiple of $\log T$. The constant does not affect the order of $T$ in the regret bound, and the concrete value is usually determined empirically, such as through cross-validation with past data. In our experimental design, as referenced in Line 1118, we chose $T_0 = 10 \log_2 T$.
>
> Re Weakness 7: Although the idea of hypothesis testing is straightforward, our innovation lies in recognizing the phase transition and finding the optimal algorithm design through injecting randomness adaptively. This optimal algorithm is further achieved through hypothesis testing.
>
> Re Weakness 8: Thanks for your suggestion. From the experimental results, when $\sigma_S^2=1$, we find that the regrets when $T=20000, 40000, 60000, 80000, 100000$ ($\log T$ = 9.90,10.60,11.00,11.29,11.51) are 220, 230, 234, 236, and 237, respectively. These points are even slightly sublinear. So, the actual performance is even better than the theoretical bound.
> When $\sigma_S^2=0$, $\log(\text{Regret})$ =   6.43, 6.75, 6.95, 7.10, 7.17 when $\log T$ = 9.90,10.60,11.00,11.29,11.51. The estimated slope by OLS is 0.47, testifying our square root $T$ regret.
> We will include this in the next version.
>
> Thank you again for your comments. We hope you find the response satisfactory. Please let us know if you have further questions.

---

> > ### Comment · Reviewer_pz8n · 2024-08-14
> >
> > I thank the authors for clarifying the details. Based on the response, I would like to raise my scores.

---

> > > ### Author Response · Authors · 2024-08-14
> > >
> > > Thanks a lot!

---

### Author Rebuttal · Authors · 2024-08-05

Thank you all for your meaningful reviews! We add a new experiment increasing the number of trajectories to 100 with various noise levels in the supply. Please see the added pdf for detailed results.

---

### Decision · Program_Chairs · 2024-09-25

**Decision:**

Accept (poster)

**Comment:**

The paper analyzes a dynamic pricing problem faced by a platform in setting its service fees, in the presence of strategic, long-living and far-sighted customers. The platform does not know the demand information.  At each time, the platform sets the service fees, and observes the resulting equilibrium prices and quantities. Using this information, the platform seeks minimize its regret against the benchmark with known demand information. The authors develop an "actions-as-instrument" approach to learn the demand, and achieve vanishing regret.

The overall consensus is that the paper is technically rigorous, and interesting, and makes a solid contribution. There were some questions about the problem motivation and connection to real world application. Given the consensus and the ratings, the paper would be a good addition to the conference program.